



# Observational assessment of the role of nocturnal residual-layer chemistry in determining daytime surface particulate nitrate concentrations

Gouri Prabhakar[1], Caroline Parworth[2], Xiaolu Zhang[1], Hwajin Kim[2,*], Dominique Young[2,^], Andreas J. Beyersdorf[3,&], Luke D. Ziemba[3], John B. Nowak[3], Timothy H. Bertram[4], Ian C. Faloona[5], Qi Zhang[2], Christopher D. Cappa[1]

[1]Department of Civil and Environmental Engineering, University of California, Davis, CA, USA
[2]Department of Environmental Toxicology, University of California, Davis, CA, USA
[3]NASA Langley Research Center, Hampton, Virginia, USA
[4]Department of Chemistry, University of Wisconsin, Madison, WI, USA
[5]Department of Land, Air and Water Resources, University of California, Davis, CA, USA
*Now at: Center for Environment, Health and Welfare Research, Korea Institute of Science and Technology, Seoul, South Korea
^Now at: Air Quality Research Center, University of California, Davis, California, USA
&Now at: Department of Chemistry, California State University, San Bernardino, CA, USA

*Correspondence to*: Christopher D. Cappa (cdcappa@ucdavis.edu), Gouri Prabhakar (gourip@ucdavis.edu)

**Abstract.** This study discusses an analysis of combined airborne and ground observations of particulate nitrate ($NO_3^-{}_{(p)}$) concentrations made during the wintertime DISCOVER-AQ study at one of the most polluted cities in the United States, Fresno, CA in the San Joaquin Valley (SJV) and focuses on development of understanding of the various processes that impact surface nitrate concentrations during pollution events. The results provide an explicit case-study illustration of how nighttime chemistry can influence daytime surface-level $NO_3^-{}_{(p)}$ concentrations, complementing previous studies in the SJV. The observations exemplify the critical role that nocturnal chemical production of $NO_3^-{}_{(p)}$ aloft in the residual layer (RL) can play in determining daytime surface-level $NO_3^-{}_{(p)}$ concentrations. Further, they indicate that nocturnal production of $NO_3^-{}_{(p)}$ in the RL, along with daytime photochemical production, can contribute substantially to the build-up and sustaining of severe pollution episodes. The exceptionally shallow nocturnal boundary layer heights characteristic of wintertime pollution events in the SJV intensifies the importance of nocturnal production aloft in the residual layer to daytime surface concentrations. The observations also demonstrate that dynamics within the RL can influence the early-morning vertical distribution of $NO_3^-{}_{(p)}$, despite low wintertime wind speeds. This





overnight reshaping of the vertical distribution above the city plays an important role in determining the net impact of nocturnal chemical production on local and regional surface-level $NO_3^-{}_{(p)}$ concentrations. Entrainment of clean free tropospheric air into the boundary layer in the afternoon is identified as an important process that reduces surface-level $NO_3^-{}_{(p)}$ and limits build-up during pollution episodes. The influence of dry deposition of $HNO_3$ gas to the surface on daytime particulate nitrate concentrations is important but limited by an excess of ammonia in the region, which leads to only a small fraction of nitrate existing in the gas-phase even during the warmer daytime. However, in late afternoon, when diminishing solar heating leads to a rapid fall in the mixed boundary layer height, the impact of surface deposition is temporarily enhanced and can lead to a substantial decline in surface-level particulate nitrate concentrations; this enhanced deposition is quickly arrested by a decrease in surface temperature, which drops the gas-phase fraction to near zero. The overall importance of enhanced late afternoon gas-phase loss to the multiday build-up of pollution events is limited by the very shallow nocturnal boundary layer. The case study here demonstrates that mixing down of $NO_3^-{}_{(p)}$ from the RL can contribute a majority of the surface-level $NO_3^-{}_{(p)}$ in the morning (here, ~80%), and a strong influence can persist into the afternoon even when photochemical production is maximum. The particular day-to-day contribution of aloft nocturnal $NO_3^-{}_{(p)}$ production to surface concentrations will depend on prevailing chemical and meteorological conditions. Although specific to the SJV, the observations and conceptual framework further developed here provide general insights into the evolution of pollution episodes in wintertime environments.

## 1 Introduction

Nocturnal processing of nitrogen oxides, $NO_x$ (= NO + $NO_2$) can strongly influence daytime air quality (Dentener and Crutzen, 1993; Brown et al., 2006c). At night, once photochemical reactions shutdown, $NO_x$ reacts with ozone ($O_3$) to form nitrate radical ($NO_3$) and dinitrogen pentoxide ($N_2O_5$) (Reactions 1 through 3a). $N_2O_5$ can react heterogeneously with airborne particles to form either nitric acid ($HNO_3$) (Reaction 4a) or, in the presence of particulate chloride, nitryl chloride ($ClNO_2$) (Reaction 4b, where $Y_{ClNO2}$ represents the molar yield of $ClNO_2$ with respect to the $N_2O_5$ reacted). In the presence of basic species like ammonia ($NH_3$), $HNO_3$ can be neutralized to form particulate nitrate ($NO_3^-{}_{(p)}$). $NO_3$ radicals can alternatively react with volatile organic compounds (VOCs), which suppresses $HNO_3$ formation (Reaction 3b). Much research has focused on the influence of nocturnal $NO_x$ processing on



the regional budgets of $NO_x$ and $O_3$ and on the oxidative capacity of the atmosphere during subsequent mornings (e.g. Brown et al., 2006b; Thornton et al., 2010; Wild et al., 2016). The corresponding impact of nighttime production of $NO_3^-{}_{(g+p)}$, a key nocturnal sink for $NO_x$, on local and regional air quality can be considerable (Lowe et al., 2015; Pusede et al., 2016) but is less often considered in detail.

$NO + O_3 \rightarrow NO_2 + O_2$ (R1)

$NO_2 + O_3 \rightarrow NO_3 + O_2$ (R2)

$NO_2 + NO_3 \leftrightarrow N_2O_5$ (R3a)

$NO_3 + VOC \rightarrow products$ (R3b)

$N_2O_5 + H_2O(het) \rightarrow 2HNO_3$ (R4a)

$N_2O_5 + Cl^-(het) \rightarrow Y_{ClNO2} + (2 - Y_{ClNO2})NO_3^-$ (R4b)

The importance of nocturnal $NO_x$ chemistry to $NO_3^-{}_{(p)}$ production can be especially important in the winter. Relative to summer, nights in winter are longer, colder and more humid, and biogenic VOC emissions tend to be smaller. This allows for a larger fraction of $NO_2$ to be oxidized to $HNO_3$ via the $N_2O_5$ hydrolysis pathway (Cabañas et al., 2001; Wagner et al., 2013) and colder temperatures favor

partitioning of nitrate to the particle-phase (Stelson and Seinfeld, 1982). In winter, nighttime $HNO_3$ production can more efficiently compete with daytime photochemically driven production due to the low photolysis rates and hydroxyl radical concentrations (Wagner et al., 2013; Pusede et al., 2016). Multiday pollution events (i.e. periods with elevated particulate matter concentrations) can occur when meteorological conditions inhibit dispersion, as is the case with persistent cold air pool formation often

found in valley regions (Whiteman et al., 2014; Baasandorj et al., 2017). During the daytime, sunlight driven convection leads to an evolution of the near-surface temperature profile and causes the atmosphere to be reasonably well mixed up to some height (typically less than 1 km; c.f. Figure S1). Radiative cooling in the late afternoon leads this mixed layer (ML) to decouple and separate into a





shallow, near surface-level nocturnal boundary layer (NBL) and a residual layer (RL) aloft, the behavior of which can be further modified by valley flows.

Nocturnal conversion of $NO_x$ to $NO_{3\ (p)}^-$ can occur either in the NBL or the RL. Surface NO emissions can substantially limit direct production of $NO_{3\ (p)}^-$ in the NBL by titrating $O_3$, depending on the initial conditions. Nocturnal surface NO emissions do not directly influence the decoupled RL, with chemical production of $NO_{3\ (p)}^-$ dependent on the $NO_x$, $O_3$ and particulate matter in the mixed layer at the time of decoupling. Box and 3D models have been previously used to assess the contribution of nocturnal processes in the RL to the daytime surface concentrations of particulate matter (PM), especially $NO_{3\ (p)}^-$ (Riemer et al., 2003; Curci et al., 2015). Yet, computational models often have difficulty in accurately predicting surface $NO_{3\ (p)}^-$ in many regions, particularly in the winter season, despite good estimations of $NO_x$ emissions (Walker et al., 2012; Terrenoire et al., 2015), although this is not always the case (e.g. Schiferl et al., 2014). Here, airborne and ground measurements made over Fresno, CA in the San Joaquin Valley (SJV) during the wintertime 2013 DISCOVER-AQ (Deriving Information on Surface Conditions from COlumn and VERtically resolved observations relevant to Air Quality; Appendix A) (Crawford and Pickering, 2014) study are used to further develop our understanding of the role that different factors play in determining surface-level $NO_{3\ (p)}^-$ concentrations.

Winters in Fresno are characterized by frequent multiday pollution episodes (Chow et al., 1999; Watson and Chow, 2002), when $PM_{2.5}$ (PM with aerodynamic diameter < 2.5 µm) mass concentrations exceed the 24-hour National Ambient Air Quality Standard (NAAQS) of 35 µg m$^{-3}$ (Figure 1). Fresno is one of the largest cities in the San Joaquin Valley (SJV), which is largely an agricultural area and suffers from some of the worst air pollution in the United States (American Lung Association, 2014). Shallow daytime mixed layer heights and low wind speeds in winter lead to the accumulation of pollutants across the valley (San Joaquin Valley Air Pollution Control District, 2003). Previous observations in the SJV region have found a build-up of $NH_4NO_3$ during pollution episodes (e.g. Chow et al., 2008). Approximately 30 − 80% of the wintertime $PM_{2.5}$ mass in this region is ammonium nitrate ($NH_4NO_3$), with a strong diurnal variability, and most other $PM_{2.5}$ being organic matter (Chow et al.,





2006; Ge, 2012; Young et al., 2016; Parworth et al., 2017). During DISCOVER-AQ specifically, $NO_3^-$ (p) was found to represent 28% of non-refractory $PM_{1.0}$ (PM with aerodynamic diameter $< 1$ µm) mass on average (Young et al., 2016).

An important role for nocturnal $NO_3^-$(p) production in this region has been previously identified based
on observations of long-term trends, the spatial and diurnal variability in $NO_3^-$(p), and the chemical environment in and around Fresno. For example, Watson and Chow (2002) reported a sharp, early morning (~9 am) increase in surface $NO_3^-$(p) concentrations on many days of a severe pollution episode in 2000 and suggested that this behavior was consistent with mixing down of nitrate-rich air from the RL aloft. Young et al. (2016) and Parworth et al. (2017) observed similar behavior more than a decade
later during DISCOVER-AQ in 2013. Pusede et al. (2016) characterized the relationship between long-term (multi-year) surface measurements of wintertime $NO_3^-$(p) and $NO_2$ in Fresno and Bakersfield and showed that the decline in $NO_3^-$(p) in SJV over time (2001-2012) was predominately driven by reduced nocturnal $NO_3^-$(p) production in the residual layer. The balance between production, especially nighttime production, and daytime losses was identified by them as critical to understanding the
multiday build-up during pollution events. Further, they concluded from DISCOVER-AQ aircraft measurements that much of the $NO_3^-$(p) production was localized over the cities given the sharp urban-rural gradients in $NO_3^-$(p); the spatial gradients in 2013 (from (Pusede et al., 2016)) seem to be sharper than gradients in 2000 (from (Chow et al., 2006)), likely reflecting the increasing localization of the $NO_3^-$(p) pollution to the urban centers as overall $NO_3^-$(p) concentrations in the region have decreased.
Brown et al. (2006a) observed that the number concentration of accumulation mode particles (0.32-1.07 µm) often increased above the surface at 90 m AGL compared to surface (7 m AGL) measurements during night, and suggested that this was due to growth of smaller particles into the accumulation mode via $NO_3^-$(p) formation. They also observed that the concentration of $NO_3^-$(p) at 90 m AGL often increased at night, suggestive of *in situ* production.

The present study builds on this literature by examining the role that aloft nocturnal nitrate production, in concert with other processes, has in determining surface $NO_3^-$(p) concentrations during the



DISCOVER-AQ campaign that took place in January and February 2013 in the SJV. Our study combines aircraft and surface observations from DISCOVER-AQ (Fig. S1). During DISCOVER-AQ, two pollution episodes were observed during which $PM_{2.5}$ concentrations were elevated (Young et al., 2016). The analysis here focuses on quantitative assessment of $NO_3^-{}_{(p)}$ concentrations during this first episode (14 – 22 January) in terms of the processes that govern the $NO_3^-{}_{(p)}$ diurnal behavior; the observed behavior during this first episode is qualitatively compared with that during the second episode (30 January – 6 February) to examine the factors that contribute to episode-to-episode variability. On flight days, *in situ* measurements of the vertical profiles of particulate and gas concentrations above Fresno (and other SJV cities) were made three times: in the mid-morning (~9:30 am), around noon and in the mid-afternoon (~2 pm). These measurements allow for assessment of the daytime evolution of the vertical distribution of PM and gases as well as characterization of the time-varying boundary layer height. They also allow for determination of the overnight evolution of the PM vertical distribution, which can be used to characterize the factors that control $NO_3^-{}_{(p)}$ concentrations in the RL. The influence of processes occurring aloft on the temporal evolution of $NO_3^-{}_{(p)}$ surface concentrations is quantitatively evaluated for this case-study using an observationally constrained 1D box model. The box model accounts for both vertical mixing (entrainment) of air to the surface and for photochemical $NO_3^-{}_{(p)}$ production, as well as $NO_3^-{}_{(p)}$ loss processes. Ultimately, the observations and analysis further illustrate how daytime surface-level $NO_3^-{}_{(p)}$ concentrations depend on a combination of both nocturnal and daytime production of $NO_3^-{}_{(p)}$, vertical mixing, differential horizontal transport in the RL overnight, daytime entrainment of clean air from the free troposphere (FT) and evaporation-driven dry deposition. The model and observations are used to examine the relative importance of these different pathways during the case-study episode considered. This work adds to the existing literature by providing an observationally based, case-study demonstration of how nocturnal processes occurring aloft—in concert with other processes—exert a major control over the evolution of pollution episodes within the SJV specifically, and likely in other regions as well.



## 2 Materials and Method

Airborne *in-situ* measurements (such as particle scattering, gas-phase concentrations, RH and temperature) during the DISCOVER-AQ campaign were made by a suite of instruments on board the P3-B NASA aircraft. The flight path flown during each of the three legs for each flight day is shown in Figure S2. The aircraft measurements were complemented by a network of ground measurement sites, of which Fresno was one. At Fresno, continuous, *in situ* measurements of the chemical composition and physical properties of particulate matter were performed along with measurement of NAAQS regulated pollutants (Young et al., 2016; Zhang et al., 2016; Parworth et al., 2017). All data are archived at the DISCOVER-AQ website (NASA Atmospheric Science Data Center). Details of all measurements made are provided in Appendix A and summarized in Table A1.

## 3 Results and Discussion

### 3.1 Vertical distribution of $NO_3^-{}_{(p)}$

The concentration and vertical distribution of $NO_3^-{}_{(p)}$ in the RL ($[NO_3^-{}_{(p)}]_{RL}$) in the morning serves as the initial condition constraint on what is mixed down to the surface as the day advances and the ML rises. Thus, knowledge of the vertical distribution of $NO_3^-{}_{(p)}$ in the RL near sunrise is needed to predict the temporal evolution of surface-level $NO_3^-{}_{(p)}$ during the daytime, as will be done below. Nighttime flights were not made during DISCOVER-AQ to allow for characterization of the overnight evolution of the RL. However, the early morning (~09:30 local time) vertical profiles over Fresno allow for characterization of the vertical structure of most of the RL near sunrise (~07:10 local time), as the surface boundary layer height at this point is still quite shallow (~50 m; see Appendix B for a description of the mixed boundary layer height determination method, Figures B1-B2). Fast measurements of total $NO_3^-$ (gas + particle, $NO_3^-{}_{(g+p)}$) were only available for a subset of flights (Pusede et al., 2016), and particulate-only $NO_3^-$ measurements were not made with sufficient time resolution to allow for robust characterization of the $NO_3^-{}_{(p)}$ vertical profile. Therefore, $NO_3^-{}_{(p)}$ vertical profiles for each flight during Episode 1 are estimated from *in situ* measurements of dry particle scattering and the influence of water uptake on scattering, i.e. from the particle hygroscopicity, and





calibrated against the slower PILS measurements (Appendix A, Figure A1). The derived, observationally constrained $NO_3^-{}_{(p)}$ profiles based on the estimated $NO_3^-{}_{(p)}$ exhibit generally good correspondence with the sparser direct measurements of $NO_3^-{}_{(g+p)}$ (Figure 2). This indicates that the estimation method is reasonable, especially since most nitrate is expected to be in the particle-phase

(Parworth et al., 2017) given the high relative total ammonium ($NH_3 + NH_4^+$) concentrations (Figure 4). Only four out of five flight days during Episode 1 have been included in this analysis due to insufficient data on 16 January.

Over Fresno, the observed afternoon (~2:30 pm) $NO_3^-{}_{(p)}$ concentrations are nearly constant with altitude up to ~400 m (the daytime boundary layer height) (Figure 2B) whereas the early-morning

$NO_3^-{}_{(p)}$ concentrations decrease steeply with altitude up to ~350 m (Figure 2A). Corresponding vertical profiles for NO, $NO_2$, $O_3$, relative humidity, temperature and total particle scattering are shown in Figures S2 (early morning) and S3 (afternoon). Like $NO_3^-{}_{(p)}$, all indicate substantial differences between the early morning and afternoon profile shapes. This provides a strong indication that altitude-specific processes occur overnight that lead to a reshaping of the $NO_3^-{}_{(p)}$ vertical profile. At some

altitudes the $NO_3^-{}_{(p)}$ in the early-morning RL is greater than the $NO_3^-{}_{(p)}$ measured in the previous afternoon, indicating net production, while at other altitudes the early-morning RL $NO_3^-{}_{(p)}$ is less than the previous afternoon, indicating net loss (Figure 2). As noted by Pusede et al. (2016), there tend to be sharp concentration gradients in $NO_3^-{}_{(p)}$ and $NO_x$ between the city and surrounding areas, with lower concentrations outside the city. Thus, whether $NO_3^-{}_{(p)}$ at a given altitude increases or decreases

overnight results from the competing effects of chemical production versus horizontal advection bringing in this (typically) cleaner air from outside the city. (In the absence of a strong jet aloft and no convective mixing, nighttime entrainment of cleaner FT air into the RL is expected to be considerably slower than horizontal advection.) Like $NO_3^-{}_{(p)}$, the boundary layer is reasonably well mixed with respect to $NO_x$, $O_3$ and particles at the time when decoupling of the RL occurs, around 3 pm the

previous day (Figure S4). Box model calculations indicate that the expected local nocturnal chemical production of nitrate in the RL should exhibit relatively minor vertical variation (due to variations in



temperature and RH) (Figure S5). In other words, without loss or dilution processes, it is expected that the $NO_3^-{}_{(p)}$ concentration would increase to a similar extent at all RL altitudes.

The substantial changes observed in the shape of the vertical profile overnight indicate that night time differential advection in the RL is a major factor in determining the shape of the morning $NO_3^-{}_{(p)}$
vertical profile during this pollution episode. Differential horizontal advection serves to export $NO_3^-{}_{(p)}$ from the urban area and import cleaner air from surrounding areas. As $NO_x$ concentrations are also lower outside of the Fresno urban area, this differential advection will also influence the over-city concentrations of precursors gases ($NO_x$ and $O_3$; Figure S3-S4) and consequently the nitrate production, with decreases likely. The important implication is that overnight advection decreases the
over-city $NO_3^-{}_{(p)}$ concentrations in the morning both directly and indirectly, which will consequently serve to limit the extent of localized pollution build-up during events. The impact of overnight differential advection on reshaping the vertical distribution of $NO_3^-{}_{(p)}$ has likely increased over the last 15 years as the sharpness of the urban-rural concentration gradients has increased (Chow et al., 2006; Pusede et al., 2016). Nonetheless, the $NO_3^-{}_{(p)}$ advected from urban areas in the RL will contribute to
the regional SJV background and serve to sustain $NO_3^-{}_{(p)}$ levels across the valley during pollution episodes.

In the summer, transport and dispersion of pollutants has been attributed to low-level winds (less than 500 m AGL) in the SJV (Bao et al., 2008). We suggest that a similar, but weaker, circulation may exist even in the winter, just at much slower wind speeds, and that this advection overnight is what leads to
differential wash out and the establishment of the particular vertical $NO_3^-{}_{(p)}$ concentration profiles in the RL. The concentration of $NO_3^-{}_{(p)}$ will likely be lowest in the early-morning RL at altitudes where horizontal advection has the greatest impact. Wind profiler measurements made in nearby Visalia, CA (65 km SE of Fresno) indicate that during the night (19:00 – 07:00) there was maximum in the mean wind speed at ~250 m, which is around the altitude at which the early-morning $NO_3^-{}_{(p)}$ concentration
is minimum (Figure 3a). Below 250 m there was a monotonic increase in the night time mean wind speed with altitude, with very slow speeds observed at the surface. Above 250 m the mean wind speed



was relatively constant to ~450 m, above which it increased with altitude. Explicit comparison between the vertical profiles of night time mean wind speed and the estimated early-morning $NO_3^-{}_{(p)}$ concentration indicates an inverse relationship ($r$ = -0.98) between the two (Figure S6). This is consistent with the idea that differential advection as a function of altitude overnight serves to shape

the early-morning concentration profiles. The wind direction at lower altitudes (~150 m) was generally more variable than those at higher altitudes (285 m or 450 m), and with a general shift from more westerly at lower altitudes (but above the surface) to more northerly near the top of the RL (Figure 3b). (Note: vector average wind speeds for each individual night were calculated and then a scalar average of these night-specific vector averages was calculated to give the episode-average mean wind

speeds. This averaging process emphasizes directional consistency of the winds on a given night, but not between nights.) The increase in $NO_3^-{}_{(p)}$ concentration at ~400 m AGL in the early-morning profile, especially noticeable on Jan 21 (Figure S7), could result from a slowing of the winds near the top of the RL or from enhanced recirculation of pollutants at higher altitudes. Regardless of reason, this work indicates that the gradient between the local (above city) and regional $NO_3^-{}_{(p)}$ and precursor gases,

evident in Pusede et al. (2016), is an important factor in determining the nighttime evolution of the RL vertical profile. Explicit characterization of the temporal evolution of the vertical structure of $NO_3^-{}_{(p)}$ within the nighttime RL would provide further insights into the altitude-specific processes that control the shape of the early-morning profile (and thus the concentration of $NO_3^-{}_{(p)}$ aloft that can be mixed to the surface in daytime).

The difference between the concentration of $NO_3^-{}_{(p)}$ at each altitude of the early morning vertical profile and that at 3 pm on the preceding afternoon ($\Delta[NO_3^-{}_{(p)}]_{RL}$) yields the net overnight $NO_3^-$ production or loss in the RL. If it is assumed that the layer with the highest $NO_3^-{}_{(p)}$ is not influenced by advection, then the $\Delta[NO_3^-{}_{(p)}]_{RL}$ in this layer provides an estimate of the maximum chemical production ($PNO_3^-$). This estimate of $PNO_3^-$ is certainly a lower bound on actual nitrate

formation given the assumption of no influence of horizontal advection, and this also does not account for produced nitrate that remains in the gas-phase (although this is likely to be small). On average, the observations indicate that chemical production overnight in the RL leads to an approximate doubling



over the initial $NO_3^-{}_{(p)}$ concentration, or 10-25 µg m$^{-3}$ of $NO_3^-{}_{(p)}$ produced over the course of the night for this episode (Table S1). Observed day-to-day variability in $PNO_3^-$ likely results from day-to-day variations in precursor ($NO_x$ and $O_3$) concentrations and $N_2O_5$ reactivity, as well as limitations of the assumption of no advection in this layer. To assess the reasonableness of this estimate of $PNO_3$ as a

maximum production rate, values of the night-specific average rate coefficients for $N_2O_5$ heterogeneous hydrolysis ($k_{N2O5}$) and associated uptake coefficients ($\gamma_{N2O5}$) needed to reproduce the observed $PNO_3^-$ are back-calculated based on the initial $NO_x$, $O_3$, and wet particle surface area and assuming $ClNO_2$ formation is negligible (see Appendix C and Table S1). The derived $k_{N2O5}$ values range from $1.3 - 5.1 \times 10^{-5}$ s$^{-1}$ with corresponding $\gamma_{N2O5}$ from $2.5 \times 10^{-4}$ to $4.8 \times 10^{-4}$. These are smaller

than values observed under water-limited conditions in other field studies (Brown et al., 2006c; Bertram et al., 2009) and lower than expected based on lab experiments (Bertram et al., 2009). $\gamma_{N2O5}$ values separately calculated from the particle composition measurements, following Bertram et al. (2009), are larger than the above back-calculated values, with $\gamma_{N2O5} \sim 10^{-3}$, and more consistent with the literature. This suggests that the $PNO_3^-$ is, in fact, a lower estimate and that the $NO_3^-{}_{(p)}$

concentration in even the lower layers of the RL is influenced by advection. Box model calculations using the (too low) back-calculated $k_{N2O5}$ and $\gamma_{N2O5}$ yield ~15-42% $NO_x$ conversion to $HNO_3$ overnight during this episode. If instead $\gamma_{N2O5} = 10^{-3}$ is used, the calculated overnight conversion is somewhat larger, ~52%. Also, if $k_{N2O5}$ and $\gamma_{N2O5}$ were assumed sufficiently large such that they are not rate limiting the overnight conversion increases further to ~63%. It should be noted that during this episode

the surface $O_3$ overnight is essentially completely titrated away by 6 pm (Figure 5). The reaction between $NO_2$ and $O_3$ (R1) is thus very slow and nighttime chemical production of $NO_3^-{}_{(p)}$ at the surface in the NBL is comparably small.

## 3.2 Vertical mixing, photochemical production and $NO_3^-{}_{(p)}$ sinks

The observed episode average surface-level $NO_3^-{}_{(p)}$ concentration exhibits a distinct, rapid increase

starting at ~ 8 am, then peaks around $10 - 11$ am local time (LT) and decreases fairly continuously after the peak, especially between $1 - 4$ pm (Figure 6A). For reference, time series of $NO_3^-{}_{(p)}$ during



the pollution episode, along with CO, NO, $NO_2$, $O_3$, temperature, surface radiation, and $PM_1$ are shown in Figure S8. Both Young et al. (2016) and Pusede et al. (2016) noted this increase, arguing it is a signature of nocturnal nitrate production. Here, we provide a more detailed examination of the specific influence of vertical mixing and nocturnal $NO_3^-{}_{(p)}$ production in the RL on the observed daytime

variability in surface-level $NO_3^-{}_{(p)}$ using an observationally constrained one dimensional box model (see Appendix D for details). In brief, the model accounts for time-dependent mixing between air in the mixed boundary layer and the RL, daytime photochemical production of nitrate, gas-particle partitioning of nitrate, entrainment of clean air from the free troposphere into the ML and loss of nitrate via dry deposition to calculate the time-dependent evolution of the surface-level $NO_3^-{}_{(p)}$ concentration.

The observed vertical profiles of $NO_3^-{}_{(p)}$ concentrations in the RL (referred to as $[NO_3^-{}_{(p)}]_{RL}$ and taken as the observed early-morning and noon profiles) provide a unique constraint for understanding and quantifying the influence of vertical mixing specifically, allowing us to expand on previous studies. The model is additionally constrained by the surface-level concentrations of $NO_2$ and $O_3$, and temporally varying ML height. The evolution of the daytime ML height and rate of entrainment are

determined using the Chemistry Land-surface Atmosphere Soil Slab (CLASS) model (https://classmodel.github.io/; Ouwersloot and Vilà-Guerau de Arellano, 2013). The CLASS model is constrained by observations of the time-dependent vertical profile measurements of temperature, RH and other gas-phase species over Fresno and by T and RH profiles and surface sensible heat flux measurements at nearby Huron, CA (~83 km SSW of Fresno) (Appendix B). Starting at around 8 am,

the ML begins to grow vertically by entraining air from the RL. It is assumed that air within the ML is instantaneously mixed throughout the volume. Within the (shrinking) RL the $NO_3^-{}_{(p)}$ is assumed to retain the initial profile shape until it reaches the maximum ML height observed in the afternoon (~12:30 pm). After this point entrainment of free tropospheric air (FT) begins. The concentration of $NO_3^-{}_{(p)}$ in FT air is determined from the vertical profile observed around noon. Photochemical

production of $HNO_3$ is calculated based on the oxidation of $NO_2$ by hydroxyl radicals, with wintertime concentrations estimated to peak around $[OH] = 10^6$ molecules $cm^{-3}$ at noon in the region (Pusede et



al., 2016). The OH concentration is assumed to scale linearly with the observed solar radiation (Figure S9).

The average calculated daytime temporal evolution of surface $NO_3^-{}_{(p)}$ from the observationally constrained box model agrees reasonably well with the average of the surface observations from the four Episode 1 flight days considered (Figure 7A). (The observed diurnal average in Figure 6 uses all of the days from Episode 1 whereas in Figure 7 only four flight days are included. This is because the initial early-morning $NO_3^-{}_{(p)}$ vertical profile is required as input to the model.) The model predictions for the individual flight days also exhibit generally good agreement with the $NO_3^-{}_{(p)}$ observations (Figure S7). Specifically, the observationally constrained model also shows a rapid increase in $NO_3^-{}_{(p)}$ beginning at 8 am, a peak around 10-11 am and a gradual, time-varying decrease through the afternoon.

Consideration of the individual processes occurring in the model demonstrates that vertical mixing down of $[NO_3^-{}_{(p)}]_{RL}$ and the shape of the $[NO_3^-{}_{(p)}]_{RL}$ vertical profile predominately control the morning-time evolution of the surface $NO_3^-{}_{(p)}$ during this episode (Figure 7 and Figure 8). The particularly steep rise in the surface-level $NO_3^-{}_{(p)}$ in the morning results from the combination of the NBL height being exceptionally shallow (only ~20 m) and the $NO_3^-{}_{(p)}$ in the low-altitude region of the RL being greater than the $NO_3^-{}_{(p)}$ in the early-morning NBL. The peak and turnover in surface-level $NO_3^-{}_{(p)}$ occurs when higher RL layers, where $[NO_3^-{}_{(p)}]_{RL} < [NO_3^-{}_{(p)}]_{ML}$, are entrained. In other words, the temporal evolution of the surface-level $NO_3^-{}_{(p)}$ is linked to the shape of the early-morning vertical $NO_3^-$ profile. Further, it should be noted that the exact model behavior is dependent on the timing of the CLASS-predicted boundary layer height increase, with the initial increase and timing of the surface-level $NO_3^-{}_{(p)}$ peak being particularly sensitive to the shape of the rise between 8 and 10 am. Nonetheless, because the NBL is so shallow here, only ~3-12% of the daytime ML height, the surface concentration is strongly impacted by the concentrations in the RL and the initial (pre-8 am) surface-level nitrate has control over daytime concentrations. Thus, the model results demonstrate that the observation of the large 10 am peak in $NO_3^-{}_{(p)}$ is a clear indication of the strong influence of nocturnal processes occurring aloft— both chemical production and advection-driven local loss—on daytime surface concentrations.



As an extreme counter-example, if there were no $NO_{3(p)}^-$ in the RL, mixing would have led to an initial decline in the early morning surface $NO_{3(p)}^-$ (Figure 8A). Alternatively, if the aloft $NO_{3(p)}^-$ concentration were assumed to be equal to that from the previous day at 3 pm (and with no vertical variability), there would not have been a sharp increase in the morning surface $NO_{3(p)}^-$ (Figure 8B). Instead, there would have been a more gradual increase from the morning into the afternoon due largely to the increasing influence of photochemical production. This is representative of a case in which there was neither aloft production of $NO_{3(p)}^-$ nor losses from advection, such that the early-morning RL concentration was determined entirely by carry-over from the prior day; in this case the difference between the early-morning surface concentration and that in the RL is small compared to the observations. If, instead, the RL $NO_{3(p)}^-$ concentration at all altitudes had been equal to the maximum $NO_{3(p)}^-$ observed in the RL (no vertical gradient in the RL), then the morning peak in surface-level $NO_{3(p)}^-$ would have occurred later and the $NO_{3(p)}^-$ concentration would be substantially higher throughout a greater fraction of the day (Figure 8C). This is representative of a case in which nocturnal production in the RL occurred, but where advection did not serve to reshape the $NO_{3(p)}^-$ vertical profile in the RL. Clearly, export of pollution from the relatively compact Fresno urban area to the broader region (and import of cleaner air) plays an important role in determining the daytime surface-level concentration of $NO_{3(p)}^-$, multi-day build up and the population exposure in this urban area. While it has previously been suggested that the morning increase in surface-level $NO_{3(p)}^-$ is indicative of mixing down of $NO_{3(p)}^-$ in the RL (Watson and Chow, 2002; Pusede et al., 2016; Young et al., 2016), the current study provides an explicit, observationally constrained demonstration of this effect and highlights the dual roles of chemical production and advective loss in the RL.

The time-evolving relative contributions of surface-level $NO_{3(p)}^-$ from the NBL, the RL and photochemical production are individually quantifiable from the model for this episode (Figure 7B). As the ML rises, the relative contribution of $NO_{3(p)}^-$ from the RL rapidly increases reaching ~80% at the 10-11 am peak. After this point, the relative contribution of $NO_{3(p)}^-$ from photochemical production increases continuously. By the time that decoupling of the NBL occurs (~3 pm), photochemically produced $NO_{3(p)}^-$ comprises 58% of surface-level $NO_{3(p)}^-$ while $NO_{3(p)}^-$ from the previous nights' RL



still comprises 40%; the contribution of $NO_3^-{}_{(p)}$ that was in the NBL is negligible (<2%). Pusede et al. (2016) showed that future decreases in $NO_x$ emissions are more likely to decrease nighttime than daytime $NO_3^-{}_{(p)}$ production. The results here therefore suggest that decreases in $NO_3^-{}_{(p)}$ may be more apparent, on average, in the morning than the afternoon since the fractional contributions of nighttime-

produced versus daytime-produced $NO_3^-{}_{(p)}$ shift throughout the day. However, care must be taken when interpreting observations from individual days since the meteorological conditions that favor observation of an early morning increase will not always occur (discussed further below). Since it is assumed here that OH scales with solar radiation, the potential for enhanced production of OH (and subsequently $NO_3^-{}_{(p)}$) in the early morning via e.g. HONO photolysis is not accounted for in the model

(Pusede et al., 2016). If this process were included, the increase in morning surface-level $NO_3^-{}_{(p)}$ would be even greater than is already calculated from mixing down of $NO_3^-{}_{(p)}$ in the RL. Since the observationally constrained model already predicts a somewhat larger peak at 10 am for surface-level $NO_3^-{}_{(p)}$ concentrations compared to the observations, early-morning photochemical production appears to have had a relatively limited influence on the morning surface-level $NO_3^-{}_{(p)}$ compared to mixing

down of nocturnal $NO_3^-{}_{(p)}$ during this episode.

While vertical mixing and the shape of the $NO_3^-{}_{(p)}$ vertical profile are what predominately drive the morning temporal evolution in the surface-level $NO_3^-{}_{(p)}$ (especially the peak) for this episode, the afternoon behavior, especially between ~1 pm and 4 pm, is shaped by the balance between photochemical production and loss via (i) dilution by entrainment of FT air and (ii) evaporation of

$NO_3^-{}_{(p)}$ and subsequent dry deposition of $HNO_3$ gas, i.e. a gas-phase pump for $NO_3^-{}_{(p)}$ loss. Here, the relative importance of these loss pathways is considered. The latter process (gas-phase pump) has been previously considered by Pusede et al. (2016) while the former (FT entrainment) was not. Loss through dry deposition of $NO_3^-{}_{(p)}$ is negligible since deposition velocities for $HNO_3$ ($v_d = 1 - 10$ cm s$^{-1}$) are much larger than for particles ($v_d = 0.001 - 0.1$ cm s$^{-1}$) (Meyers et al., 1989; Horii et al., 2005; Farmer

et al., 2013; Pusede et al., 2016). These loss mechanisms ultimately limit the extent of the pollution episode build up. Once the daytime model ML reaches maximum height entrainment into the ML of



typically cleaner air from just above the ML (i.e. from the FT) occurs. The time-evolving entrainment rates are estimated from the CLASS model (Appendix C).

Considering the gas-phase pump, the warm (typically 290 K) and dry (RH = 40% or less during the campaign) afternoon conditions enhance evaporation of $NO_3^-{}_{(p)}$ relative to night time and early

morning conditions, thereby increasing loss through dry deposition of $HNO_3$ gas in the afternoon (Pusede et al., 2016). However, total ammonia is in substantial excess (3.8 – 8.9 times $NO_3^-{}_{(g+p)}$ on a molar basis), with thermodynamic calculations indicating that the gas-phase fraction of $NO_3^-$ is <0.15 during the daytime and near zero at night when it is colder and RH is higher (Figure 4). These estimates of the gas-phase fraction of $NO_3^-$ are similar to the observational measurements of Parworth et al.

(2017), who determined the daytime and nighttime averages during the first episode were $0.08 \pm 0.03$ ($1\sigma$) and $0.04 \pm 0.05$ ($1\sigma$), respectively. Importantly, the gas-phase fraction here is substantially smaller than that estimated in Pusede et al. (2016) who found a daytime gas-phase fraction of 0.4 (median) and a 24-h average of 0.15. Consequently, loss of nitrate via the gas-phase pump is less than in their analysis and suggests that the role of this pathway was likely overestimated. The general

influence of the gas-phase fraction on loss via dry deposition is shown in Figure S10. In general, the results indicate that the gas-phase fraction has a strong influence on the loss of $NO_3^-{}_{(p)}$ due to $HNO_3$ deposition.

Including both FT entrainment and dry deposition, the box model reasonably reproduces the observed afternoon decrease in surface-level $NO_3^-{}_{(p)}$. This allows assessment of the relative importance of these

two loss processes by turning them off one at a time (Figure 7C). The calculations indicate that entrainment of clean FT air plays an important role in the afternoon surface concentration decline. Without entrainment, the model predicts that the afternoon $NO_3^-{}_{(p)}$ would be ~18% higher, leading to a double-humped daytime profile. Despite the relatively low gas-phase fraction, the gas-phase pump also contributes to the afternoon decline. The model results indicate that these two loss processes

contribute approximately equally to the afternoon decline. There are, however, a few hours when the gas-phase pump is potentially of extreme importance. When the RL decouples and the surface mixed



layer becomes quite shallow the rate of loss due to dry deposition is enhanced. This leads to a rapid decrease in surface-level $NO_3^-{}_{(p)}$. Yet, the concurrent decrease in the NBL temperature and increase in RH and $NH_3$ enhances partitioning of nitrate to the particle-phase, thereby limiting the impact of this rapid decline over time. (In the model here, the decoupling is assumed to occur very rapidly while the temperature and RH changes are from observations and occur more gradually. If the decoupling was actually slower the influence of the gas-phase pump at this point in time would be reduced and the modelled decrease in $NO_3^-{}_{(p)}$ that occurs around 3-5 pm would be less than shown.)

The model predicts that after decoupling and cooling occur the surface-level $NO_3^-{}_{(p)}$ will continue to decrease at ~2% h$^{-1}$ overnight via the gas-phase pump, which is similar to the loss rate observed between midnight and 7 am (Figure 7A). If the gas-phase pump is turned off completely (i.e. the nitric acid deposition velocity is set to zero) there is an increase in the modelled $NO_3^-{}_{(p)}$ that begins at ~3 pm (when decoupling occurs) and continues until 6 pm (Figure 7C). This is a result of the continual decrease in temperature and increase in RH enhancing partitioning to the particle-phase. Although not a focus of this study, on some days, there is a sharp increase in surface-level $NO_3^-{}_{(p)}$ observed in the evening, starting around 8 pm (LT). While this could theoretically result from enhanced partitioning to the particle-phase at night, the timing does not match the observed temperature and RH variations. Surface-level chemical production of nitrate via $N_2O_5$ hydrolysis could alternatively be the source of this increase, but given the near-zero surface-level $O_3$ concentration due to titration by NO the production via this pathway would be insufficient. This evening increase is observed on many days, although with somewhat variable timing and magnitude (Figure 8). Thus, it may be that the evening increase results from advection to the measurement site of air from a not-to-distant location (given low wind speeds) that has higher surface concentrations. Regardless, while the reason for this night time increase in surface $NO_3^-{}_{(p)}$ remains unclear, the occurrence does not impact the analysis of the early-morning and daytime $NO_3^-{}_{(p)}$ behavior.

The cumulative impact of the nocturnal production in the RL, daytime photochemical production and afternoon loss processes is that the $NO_3^-{}_{(p)}$ concentration at ~3 pm, the point when decoupling of the



RL occurs, is slightly higher than that at 8 am during the episode. Therefore, there is a gradual net increase (average of 1.32 µg m$^{-3}$ day$^{-1}$) in surface-level NO$_3^-$$_{(p)}$ as the episode progresses, albeit with day-to-day variability (Figure 6B). While decreasing NO$_x$ emissions and NO$_3^-$$_{(p)}$ production, especially nocturnal production (Pusede et al., 2016), is the most direct and reliable route towards decreasing

surface NO$_3^-$$_{(p)}$ concentrations (Kleeman et al., 2005), decreases in NH$_3$ could theoretically also have some influence on NO$_3^-$$_{(p)}$ by increasing the efficiency of the gas-phase pump. However, this will only be the case if NH$_3$ decreases exceed decreases in NO$_x$ by at least a factor of five such that the ratio between the two is changed substantially and the gas-phase fraction is increased (Figure 4). Such preferential targeting of NH$_3$ sources is therefore highly unlikely to be an efficient control strategy, at

least for the SJV where the total ammonia-to-nitrate ratio is large. In regions where the NH$_4^+$$_{(g+p)}$:NO$_3^-$$_{(g+p)}$ molar ratio is closer to unity, the nitrate partitioning is more sensitive to changes in this ratio and thus ammonia control could potentially prove effective.

## 3.3 Comparison between episodes

The above analysis focuses on observations made during one pollution episode, but there was a second

pollution episode observed during DISCOVER-AQ (Jan. 30-Feb. 5, 2013). The episode-averaged diurnal behavior of the surface NO$_3^-$$_{(p)}$ concentration for this second episode showed evidence of an early morning increase, but the increase is not as sharp as the first episode (Figure 9). Additionally, the day-to-day variability in the surface NO$_3^-$$_{(p)}$ was much greater during the second episode; on some days, there was minimal evidence of an early-morning increase but on others there was a substantial

increase. The shapes of the early morning vertical NO$_3^-$$_{(p)}$ profiles (around 9:30 am) were notably different during Episode 2 on two of the flight days as well, as was the evolution of the profiles from morning to afternoon (Figure S11). The afternoon mixed layer heights were much higher during Episode 2 than Episode 1, ranging from 600-700 m AGL compared to 300-400 m AGL, respectively. The early-morning mixed layer heights were also higher during Episode 2 (~170 m) compared to

Episode 1 (around 70 m). During Episode 1, the surface-level winds exhibit a consistent shift in direction from easterly in the early morning (5-8 am) to southerly in the later morning (9 am-12 pm), and the mean surface-level wind speed increased over this same period, from 0.31 m s$^{-1}$ to 0.82 m s$^{-1}$





(Figure 9). In contrast, during Episode 2 there was a lack of day-to-day consistency in the surface wind direction, especially during the early morning (5-8 am), and there was a more substantial change in the mean surface-level wind speed from early morning to later morning, from 0.32 m s$^{-1}$ to 1.12 m s$^{-1}$ (Figure 9). The Episode 2 mean night time aloft wind speeds were also overall lower and more constant

with altitude, with little variability from 150 m to 400 m, although still with a substantial increase from the surface (Figure S12). The aloft nocturnal winds during Episode 2 were somewhat more variable than Episode 1 winds in terms of the wind direction (Figure 3 versus Figure S12).

Overall, this increased day-to-day variability in both the surface $NO_3^-{}_{(p)}$ and wind behavior, and a difference in the evolution of the $NO_3^-{}_{(p)}$ vertical profiles from early morning to late morning/early

afternoon in Episode 2 compared to Episode 1, suggests that the meteorological conditions during the second episode were generally less conducive to simple interpretation using the mixing model discussed above. Instead, it seems that advection and export from the urban area were of increased importance during Episode 2, both overnight and especially in the early-to-mid morning. The contrasting behavior between the two episodes suggests that while the observation of a sharp, early-

morning rise and peak in surface-level $NO_3^-{}_{(p)}$ (such as during the first episode) might be generally considered a strong indicator of the production of $NO_3^-{}_{(p)}$ in the RL, the absence of such a feature does not preclude an important role for nocturnal production aloft.

## 4 Conclusion

This work combines surface and aircraft observations made during a pollution episode in 2013 to

demonstrate that in the San Joaquin Valley (specifically Fresno, CA) production of $NO_3^-{}_{(g+p)}$ in the nocturnal residual layer can play a crucial role in determining daytime surface concentrations of particulate $NO_3^-$ in winter, when photochemical production is relatively slow and morning boundary layers are extremely shallow. The influence of processes occurring in the aloft RL on $NO_3^-{}_{(p)}$ surface concentrations is evident in the $NO_3^-{}_{(p)}$ diurnal variability, specifically the occurrence of a mid-

morning peak in surface-level $NO_3^-{}_{(p)}$. While the mid-morning peak has been previously suggested as a signature of nocturnal nitrate production aloft (Watson and Chow, 2002; Brown et al., 2006a; Pusede



et al., 2016; Young et al., 2016), the current study makes novel use of vertical profiles of $NO_3^-{}_{(p)}$ concentrations measured multiple times on individual days to quantitatively illustrate the importance of nocturnal processes on surface concentrations. The analysis shows that the $NO_3^-{}_{(p)}$ concentration in the morning-time mixed boundary layer can be dominated by nocturnally produced $NO_3^-{}_{(p)}$; vertical

mixing has a particularly large impact on the surface concentrations here due to the nocturnal boundary layer being exceptionally shallow. In the afternoon, photochemically produced nitrate contributes the majority of the total $NO_3^-{}_{(p)}$ burden for the episode examined, but still with a substantial contribution from nocturnal production. The case-study here illustrates that nocturnal $NO_3^-{}_{(p)}$ production can play a critically important role in the build-up and sustaining of pollution episodes in the SJV, supporting

previous suggestions made, in part, on the basis of calculated chemical production values and an assessment of multi-year trends in the relationship between $NO_3^-{}_{(p)}$ and $NO_2$ (Pusede et al., 2016). Production of $NO_3^-{}_{(p)}$ in the RL can vary widely based on initial concentrations of its precursor gases, as well as the rate of heterogeneous uptake of $N_2O_5$ by particles. It may be that production of $NO_3^-{}_{(p)}$ via the $N_2O_5$ hydrolysis pathway may be significant in the aloft RL in other regions with similar

geographical and meteorological conditions, such as Salt Lake Valley, Utah (Kuprov et al., 2014; Baasandorj et al., 2017). However, in valley regions with lower $NO_x$ or $O_3$ the $P$NO$_3^-$ may be lower, thus limiting the importance of this pathway (Akira et al., 2005; Bigi et al., 2012).

The current work also demonstrates that a difference exists between the shape of the typical vertical profiles of $NO_3^-{}_{(p)}$ in afternoon and early-morning over Fresno. This difference is shown to very likely

result from altitude-specific horizontal advection in the nocturnal RL leading to differential wash-out of $NO_3^-{}_{(p)}$ and precursor gases, rather than from differences in chemical production rates. Consequently, there is a steep vertical gradient in $NO_3^-{}_{(p)}$ in the early-morning RL that, in turn, influences the temporal evolution of surface-level $NO_3^-{}_{(p)}$ during the day, especially in early morning. Ultimately, differential advection is shown to have an important role in limiting the maximum surface-

level concentration of $NO_3^-{}_{(p)}$ observed within the urban area during the day, a result of the urban-rural gradients being particularly steep (Pusede et al., 2016). Absent this overnight export of pollution from the city, nitrate pollution would build up during pollution events to a much greater extent. However,



advection likely contributes to the build-up of $NO_{3(p)}^-$ throughout the valley, outside of the cities. Daytime loss processes are also shown to help limit the multi-day build-up of surface-level $NO_{3(p)}^-$. Afternoon entrainment of air from the cleaner free troposphere into the ML (and export of mixed-layer air to the FT) is shown to be an important loss process for particulate nitrate. Janssen et al. (2012; 2013) have similarly identified afternoon loss via FT entrainment as an important process shaping the diurnal variability of surface-level organic aerosol concentrations in forested areas that are dominated by organic aerosol. Loss of $NO_{3(p)}^-$ via dry deposition of $HNO_3$ and subsequent evaporation of $NH_4NO_3$ is found to contribute to afternoon particulate nitrate loss, but the effect is limited by the (relatively) high afternoon boundary layer and the small gas-phase fraction of nitrate (<0.15). However, this gas-phase pump may have a substantial influence on the surface concentrations in the few hours just after decoupling of the RL occurs, when the boundary layer height is low and it is still sufficiently warm. Consistent with previous suggestions (Kleeman et al., 2005; Pusede et al., 2016), we conclude that control strategies for the region should focus on reduction of concentrations of $NO_x$ and $O_3$ (the latter of which might require VOC controls) in the mid-afternoon, specifically around the time that the RL decouples from the surface layer, as this largely determines the production rate of nitrate in the aloft RL.

**Appendix A: Measurements**

**A1 Airborne Measurements**

Airborne measurements used in this paper were made from the P3-B NASA aircraft during the DISCOVER-AQ field campaign in January-February, 2013 in San Joaquin Valley (SJV), California. All data are available from the publicly accessible DISCOVER-AQ website (NASA Atmospheric Science Data Center).

The P3-B was equipped with an array of instruments to measure both gas and particle-phase properties. A TSI-3563 nephelometer provided total scattering from dry particles at 450, 550 and 700 nm and scattering at 550 nm by particles at 80% RH (Beyersdorf et al., 2016). Gas-phase $NH_3$ was measured





using a cavity ringdown spectroscopy with a Picarro G2103 (von Bobrutzki et al., 2010), using the NOAA aircraft $NH_3$ inlet and calibration scheme as in Nowak et al. (2010). Measurements of NO, $NO_2$, $NO_x$, and $O_3$ were obtained through a 4-channel chemiluminiscence instrument (Brent et al., 2015). CO and $CH_4$ were measured with a differential absorption CO measurement spectrometer

(Sachse et al., 1987). Total gas ($HNO_3$) + particle ($NH_4NO_3$) nitrate were measured using thermal dissociation – laser induced fluorescence (TD-LIF), where $HNO_3$ and volatilizable particulate nitrate are converted into $NO_2$ for detection (Day et al., 2002). While the TD-LIF instrument is not optimized for particle sampling, most of the particulate mass was in the submicron size range and thus inertial losses will likely only lead to a small (if any) negative bias in the measured particulate nitrate (Pusede

et al., 2016). Aerosol size distributions for $0.06 - 1.0$ µm diameter particles were measured with an ultra-high sensitivity aerosol spectrometer (UHSAS). The UHSAS uses an optical sizing method, but is calibrated relative to mobility diameter. The P3-B flew throughout the SJV on 10 days and performed vertical spirals over six sites across the valley, including Fresno. The location of these sites and the flight path are shown in Figure S2. This same flight path was repeated three times every day

between approximately 8:30 am and 3:00 pm, with vertical profiles over Fresno at approximately 9:30-10:00 am, 12-12:30 pm and 2:30-3:00 pm. This enables assessment of the evolution of the species-specific vertical profile during the day across the valley. Out of the ten research flights during the campaign, only eight of them have been used here due to gaps in the dataset. Four of these days are during the first pollution episode (Jan. 18, 20, 21 and 22) and four are during the second pollution

episode (Jan. 30 and 31, and Feb. 1 and 4).

Observations of the light scattering coefficient at 550 nm ($\sigma_{sca}$) for dry and humidified particles (no size cut-off) made from the P3-B (Beyersdorf et al., 2016) have been used to estimate the vertical distribution of PM mass and $NO_3^-{}_{(p)}$ concentrations. Scattering is linearly related to the total mass concentration of PM. The observed hygroscopicity is dependent on particle composition, with higher

hygroscopicity indicative of a higher particulate inorganic fraction and lower hygroscopicity indicative of a higher particulate organic fraction; the relationship between hygroscopicity and the inorganic fraction (or the organic fraction) is reasonably linear when the inorganic species are primarily



ammonium sulfate and ammonium nitrate (Zhang et al., 2014), as these have similar hygroscopicities (Petters and Kreidenweis, 2007). The particulate nitrate concentration is much larger than the particulate sulfate concentration, as determined from both the surface and aircraft measurements (< 600 m AGL), with nitrate-to-sulfate mass ratios of 8 and 16, respectively (both determined from PILS

measurements). Thus, the observed hygroscopicity is primarily reflective of the particulate nitrate fraction (Parworth et al., 2017). More specifically, a linear relationship was observed between surface-level measurements of dry $\sigma_{scat}$ and $PM_{1.0}$ (= black carbon (BC) + non-refractory $PM_{1.0}$, ($NR$-$PM_1$)) mass concentrations in Fresno (slope = 2.83 $m^2$ $g^{-1}$ with intercept forced through zero; Figure A1a). Only data points between 8 am and 4 pm were included in determining this relationship to reflect the

time period during which the airborne measurements were obtained. The observed relationship for dry, surface-level $\sigma_{sca}$ and $NR$-$PM_1$ is used to estimate the $NR$-$PM_1$ concentration during the vertical profiles from the aircraft dry $\sigma_{scat}$ measurements. The hygroscopicity (water uptake) of a particle depends on its chemical composition. Inorganic components, predominantly $NO_3^-$ and ammonium in the wintertime SJV region (Young et al., 2016), are highly hygroscopic while organic components of

$PM_1$ tend to have much lower hygroscopicity (Petters and Kreidenweis, 2007). Thus, measurements of the particle hygroscopicity can be used to estimate the ratio of inorganic to organic mass in the sampled PM (Massoli et al., 2009; Parworth et al., 2017). The average particle hygroscopicity was characterized by the optical hygroscopicity parameter, $\gamma$ defined by Equation A1.

$$\gamma = \frac{\ln\left[\frac{\sigma_{sca,wet}}{\sigma_{sca,dry}}\right]}{\ln\left[\frac{100-RH_{dry}}{100-RH_{wet}}\right]} \qquad (A1)$$

where $\sigma_{scat,wet}$ and $\sigma_{scat,dry}$ are the scattering coefficients (in $Mm^{-1}$) measured under wet ($RH_{wet}$ = 80%) and dry ($RH_{dry}$ = 20%) conditions respectively. The parameter $\gamma$ varies reasonably linearly with the particle inorganic mass fraction (Massoli et al., 2009). Therefore, an initial estimate of $NO_3^-{}_{(p)}$ concentrations at high time resolution, and thus as a function of altitude, is obtained from the equation $[NO_3^-{}_{(p)}] = \gamma \cdot \sigma_{sca,dry}/2.83$. The factor of 2.83 has units of $m^2$ $g^{-1}$ and comes from the $\sigma_{scat}$ versus $NR$-

$PM_1$ relationship determined above. However, previous studies show some variability in the linear



relationship between $\gamma$ and inorganic mass fraction and, importantly, typically have slopes somewhat less than unity and non-zero intercepts, as is assumed in the above conversion (e.g. Massoli et al., 2009). Therefore, the low-time-resolution aircraft PILS $NO_3^-{}_{(p)}$ measurements (which are not appropriate for vertical profiles) were used to calibrate the above high-time-resolution $NO_3^-{}_{(p)}$

estimates. There was a strong, linear correlation between the $NO_3^-{}_{(p)}$ observed by the PILS and the initially estimated $NO_3^-{}_{(p)}$ (Figure A1b). This demonstrates the general validity of the estimation approach. However, the PILS $NO_3^-{}_{(p)}$ concentrations were, on average, 22% lower than the initially estimated $NO_3^-{}_{(p)}$. Therefore, the initially estimated $NO_3^-{}_{(p)}$ concentrations were adjusted downwards by 22%, and the final expression relating $\sigma_{sca,dry}$ (in $Mm^{-1}$) and $\gamma$ to $NO_3^-{}_{(p)}$ concentrations (in $\mu g\ m^{-3}$)

is:

$$[NO_3^-{}_{(p)}] = \frac{\gamma \cdot \sigma_{sca,dry}}{3.63} \tag{A2}$$

The uncertainty in the estimated $[NO_3^-{}_{(p)}]$ is approximately 20%, based on the scatter around the best-fit line in Figure A1.

**A2 Ground Measurements**

Fresno (36.745 °N, 119.77 °W) was a "supersite" where comprehensive, continuous measurements of the chemical and physical properties of particulate matter were made. Chemical composition of non-refractory $PM_{1.0}$ was measured by a High Resolution Time-of-Flight-Aerosol Mass Spectrometer (HR-ToF-AMS) (Young et al., 2016). The soluble fraction of $PM_{3.0}$ was characterized using a Particle-Into-Liquid Sampler (PILS) coupled to an ion chromatograph (Parworth et al., 2017). Gas-phase water-

soluble species were collected at 5 - 7 hr time resolution using an automatic-switching annular denuder system placed in front of the PILS and were analyzed offline with ion chromatography after extraction (Parworth et al., 2017). The combination of the denuder measurements and the particle measurements allowed for determination of the gas-phase fraction of nitrate. Light extinction and light absorption coefficients were measured using the UC Davis cavity ringdown-photoacoustic spectrometer, and

scattering coefficients were determined by difference (Cappa et al., 2012; Lack et al., 2012). Refractory



black carbon concentrations were measured using a single particle soot photometer (Schwarz et al., 2006). *In situ* gas-phase measurements of NO, $NO_2$ and $O_3$, along with environmental factors (T and RH) were made by the California Air Resources Board. Particle size distributions were measured using a Scanning Mobility Particle Sizer (SMPS; size range: $10 - 800$ nm), and an Aerodynamic Particle Sizer (APS; $700$ nm $- 6$ μm). Measurements included in the current study are listed in Table A.

Additionally, a radiosonde was used to obtain vertical profiles of pressure, temperature and humidity over nearby Huron (36.203 °N, 120.103 °W) twice a day, once in the morning around 8 AM and again in the evening 4 PM. Measurements of wind speed and wind direction as a function of altitude at nearby Visalia, CA are from the National Oceanic and Atmospheric Administration (NOAA) Profiler Network (https://www.esrl.noaa.gov/psd/data/obs/instruments/WindProfilerDescription.html).

## Appendix B: Determining Mixed Boundary Layer Height

The mixed layer (ML) heights have been determined from each of the vertical profiles of potential temperature (θ), relative humidity (RH), CO and $CH_4$ measured from the P3-B aircraft. Example profiles for each of the three flight legs on 18 January 2013 are shown in Fig. B1. The altitude at which there is a strong change in the slope, from approximately altitude-independent to having a steep gradient, is determined to be the top of the ML. The vertical profile measurements allow for determination of the ML height over Fresno around 10:00 am, 12:30 pm and 2:30 pm. The ML height at 8 am is separately determined from the radiosonde measurements at nearby Huron (located 83 km SSW), as the flight data do not allow for characterization of ML height this early. It is assumed that the 8 am ML measurements at Huron are representative of the ML heights in Fresno. The observed ML height increases with time from 8 am until approximately noon or 1 pm, after which it is approximately constant. The rise in ML height with time is modelled using the Chemistry Land-surface Atmosphere Soil Slab (CLASS) model (Vilà-Guerau De Arellano, 2015). The CLASS model allows for estimation of ML heights with finer time resolution than the observations (i.e. in between flights; shown as black dots in Figure B2) and of the corresponding time-dependent entrainment velocities.



The model input parameters are constrained by observations from nearby Huron of the nocturnal boundary layer height, the morning inversion strength (~ 8 am), the sensible surface heat flux, the friction velocity, and the lapse rate through the residual layer, as well as by an estimate of the subsidence rate based on Trousdell et al. (2016). The model inputs are adjusted to ensure that the

modelled ML growth agrees reasonably well with the observations from the P3-B over Fresno (Figure B2). The resulting average entrainment velocities in the afternoon (1 – 4 pm) from the CLASS model agree well with independently determined entrainment rates based on afternoon decline in $SO_4^{2-}{}_{(p)}$ for the Episode 1 days. Since $SO_4^{2-}{}_{(p)}$ is effectively non-volatile and since photochemical production via oxidation of $SO_2$ is relatively slow, the decline in $SO_4^{-}{}_{(p)}$ in the afternoon can be attributed solely to

dilution from entrainment of "clean" FT air since the influence of the gas-phase pump is small. After 3 pm the boundary layer is assumed to linearly drop over a 1-hour period to the NBL height observed at 8 am the same day.

The sensitivity of the box model to the boundary layer growth predicted by the (observationally constrained) CLASS model has been examined. An alternative boundary layer growth profile was

estimated by fitting the observed P3-B ML heights using a sigmoidal function (Figure B3). The general shapes of the CLASS and sigmoidal profiles are similar, although the sigmoidal profile exhibits a somewhat faster rise. Entrainment of FT air in the afternoon for the sigmoidal growth profile was accounted for using the average entrainment rates estimated from the observed $SO_4^{2-}{}_{(p)}$ loss rates and assuming that entrainment begins at noon, when the BL height was near the maximum. The same linear

decrease in the BL height starting at 3 pm was assumed. The use of this alternative model yields a diurnal $NO_3^{-}{}_{(p)}$ profile for Episode 1 that is very similar to that obtained using the CLASS model (Figure B3). This indicates that the general behavior of the diurnal surface $NO_3^{-}{}_{(p)}$ profile is not particularly sensitive to the treatment of the BL rise and that the results obtained here are robust.

**Appendix C: Nocturnal Reactions in the RL**

**C1 $N_2O_5$ production and heterogeneous reactivity**



The gas-phase and heterogeneous chemistry occurring in the RL was assumed to follow the reaction scheme indicated by Reactions 1-4. Focusing first on the heterogeneous hydrolysis of $N_2O_5$, one estimate of the night-specific average rate coefficients for $N_2O_5$ heterogeneous hydrolysis ($k_{N2O5}$) is obtained through consideration of the initial concentrations of precursor gases and the observed

maximum overnight increase in the RL $NO_3^-{}_{(p)}$, $PNO_3^-$. More specifically, a 1D box model including nocturnal gas-phase chemistry and heterogeneous reaction of $N_2O_5$ with particles was run iteratively to determine an average $k_{N2O5}$ for the night (19:00-08:00; 13 hours) such that it reproduced the observed $PNO_3^-$. The observed chloride at Fresno was small (1% of $PM_{1.0}$) during the episode, and thus formation of nitryl chloride ($ClNO_2$) can be reasonably neglected (Young et al., 2016). Since, the

boundary layer is fairly well-mixed in the afternoon, surface-level observations of $NO_x$, $O_3$, $NO_3^-{}_{(p)}$, particle wet surface area and temperature at 3 pm on the preceding day were used as initial conditions. Based on back-calculated $k_{N2O5}$ values, night-specific values of heterogeneous $N_2O_5$ uptake coefficient ($\gamma_{N2O5}$) were determined from:

$$k_{N2O5} = \frac{\omega \cdot S_a \cdot \gamma_{N2O5}}{4} \tag{C1}$$

where $\omega$ is the mean molecular speed of $N_2O_5$ (256 m s$^{-1}$), $S_a$ is wet particle surface area, and $\gamma$ is the $N_2O_5$ heterogeneous uptake coefficient (Brown et al., 2006c). The wet particle surface area was calculated from the observed dry particle size distributions, particle hygroscopicity and RH. The resulting back-calculated $k_{N2O5}$ values from Eqn. C1 were in the range $1.3 - 5.1$ x $10^{-5}$ s$^{-1}$. The corresponding back-calculated $\gamma_{N2O5}$ were in the range $2.5$ x $10^{-4}$ to $4.8$ x $10^{-4}$ (Table S1), which as

noted in the main text are somewhat smaller than values observed under water-limited conditions in other field studies and lower than expected based on lab experiments (Bertram et al., 2009).

A second estimate of the $\gamma_{N2O5}$ values is calculated from the particle composition following Bertram et al. (2009). The calculated $\gamma_{N2O5}$ depend on the particle water content (specifically, the $[H_2O]/[NO_3^-{}_{(p)}]$ and thus RH) and the chloride fraction. The composition-calculated $\gamma_{N2O5}$ ($\sim 10^{-3}$) are larger than the

above back-calculated values and more consistent with the literature although on the lower side of



previous measurements (Brown et al., 2006c; Bertram et al., 2009). That the back-calculated $\gamma_{N2O5}$ are smaller than the $\gamma_{N2O5}$ calculated from the composition is likely a consequence of the $P$NO$_3^-$ being an under-estimate relative to the true overnight production in the RL. This is because the observed $P$NO$_3^-$ is taken as the difference between the previous afternoon and early morning NO$_3^-$$_{(p)}$

concentration in the aloft RL layer having the maximum morning concentration. This does not account for the influence of advection, which is most likely going to reduce the morning NO$_3^-$$_{(p)}$ relative to if there were no advection.

## C2 Reactions with VOCs

Not considered in the above is the reaction of the NO$_3$ radical with VOCs. NO$_3$ radicals react rapidly

with alkenes and more slowly with alkanes and other species. NO$_3$ reaction with VOCs can lead to hydrogen abstraction and direct formation of HNO$_3$, especially for reactions with alkanes. For alkenes and aromatics, NO$_3$ reaction typically proceeds via NO$_3$ addition and formation of organic nitrates. The latter would suppress formation of particulate inorganic nitrate but can serve as an important source of particulate organic nitrate (Kiendler-Scharr et al., 2016). Organic nitrate formation has been

observed as an important source of summertime organic aerosol in Bakersfield, CA (also in the SJV) (Rollins et al., 2012). VOC concentrations and reactivity are likely much lower during the colder winter compared to the warmer summer (Pusede et al., 2014), and thus reaction of VOCs with NO$_3$ radicals is likely much suppressed.

Concentrations of a broad suite of VOCs were measured via whole air canister sampling during

DISCOVER-AQ. These measurements can be used to assess the potential influence of NO$_3$ reaction with VOCs on HNO$_3$ and NO$_3^-$$_{(p)}$ formation. The nitrate reactivity towards each VOC is calculated as $k_{VOC+NO3}\cdot$[VOC], where the $k_{VOC+NO3}$ is the VOC-specific rate coefficient and [VOC] is the VOC concentration (Ng et al., 2017). Average afternoon VOC concentrations are used, which should be representative of the initial concentrations in the RL. The VOCs are ranked according to their reactivity

with NO$_3$. The top 20 VOCs are considered explicitly, and all other VOCs are lumped into a common VOC species with the average concentration and $k_{rxn}$ of these species (Table S2). Reactions between





NO$_3$ and alkenes and aromatics are assumed to form (unreactive) organic nitrates while reactions between NO$_3$ and all other species are assumed to form HNO$_3$ and an organic product species. The influence of NO$_3$ reaction with VOCs on NO$_3^-$$_{(p)}$ is assessed by calculating the overnight production of HNO$_3$ both with and without VOCs using typical afternoon NO (3 ppb), NO$_2$ (20 ppb) and O$_3$ (27 ppb) concentrations and for $k_{N2O5}$ ranging from 1 x 10$^{-5}$ s$^{-1}$ to 3 x 10$^{-4}$ s$^{-1}$. HNO$_3$ produced from N$_2$O$_5$ hydrolysis is tracked separately from HNO$_3$ produced from NO$_3$ reaction with VOCs. The HNO$_3$ production via N$_2$O$_5$ hydrolysis decreases marginally when VOC reactions are included. The HNO$_3$ suppression ranges from ~12% for $k_{N2O5}$ = 10$^{-5}$ s$^{-1}$ to 5% for $k_{N2O5}$ = 10$^{-4}$ s$^{-1}$ (Figure D1). However, the calculations indicate that much of this HNO$_3$ suppression is potentially offset by HNO$_3$ production from reaction of NO$_3$ with non-alkene or aromatic compounds. For larger $k_{N2O5}$ the net suppression is only 5%, with the suppression decreasing as $k_{N2O5}$ decreases. At the lowest $k_{N2O5}$ (10$^{-5}$ s$^{-1}$) the calculations indicate that the inclusion of the NO$_3$ + VOC reaction actually leads to an increase in the net HNO$_3$ production (Figure D1). Overall, these calculations suggest that reaction of NO$_3$ with VOCs has a relatively minor influence on the overnight local production of HNO$_3$ in the RL.

**Appendix D: Box Model Details**

The box model for calculating the time-varying surface concentrations of NO$_3^-$$_{(p)}$ accounts for: (i) mixing of air in the aloft RL with the surface air, including the time-dependent rise and fall of the boundary layer; (ii) daytime photochemical production of HNO$_3$ from the OH + NO$_2$ reaction; (iii) T- and RH-dependent gas-particle partitioning of ammonium nitrate; (iv) afternoon entrainment of air from the free troposphere; (iv) competition between condensation of HNO$_3$ onto existing suspended particles versus loss via dry deposition; (v) dry deposition of particulate NO$_3^-$$_{(p)}$. The kinetic equations were solved in the data analysis program Igor (Wavemetrics) and set up using the kinsim Igor package, developed by Harold Stark (http://www.igorexchange.com/node/1333). The model was initialized with the observed NO$_3^-$$_{(p)}$ measured by the AMS at surface-level at 12 am and run in 10 minute steps. For each time step, the photochemical production equations used the instantaneous observed NO$_2$ and temperature, and estimated OH concentration and ML height. The fraction of NO$_3^-$ in gas-phase, $f$, for





each time step was determined based on the instantaneous conditions using the chemical thermodynamic model, ISORROPIA II in the forward mode, with the phase state set as metastable (Fountoukis and Nenes, 2007). ISORROPIA was initialized with the observed particulate composition, specifically $NO_3^-$ and $SO_4^{2-}$ (AMS), and $Na^+$, $K^+$ and $Cl^-$ (PILS). (The PILS and AMS sampled particles

of somewhat different size, with the PILS sampling $PM_3$ and the AMS $PM_1$. The AMS observations are available at higher time-resolution, and thus preferable to use here. However, there are challenges in quantitative characterization of $Na^+$, $K^+$ and $Cl^-$ using the AMS, and thus the PILS was used instead for these species. Comparison of the AMS and PILS $NO_3^-$ and $SO_4^{2-}$ indicates that the AMS-measured concentrations are ~10% lower than for the PILS, attributable to mass between 1 and 3 µm (Parworth

et al., 2017). The $Na^+$, $K^+$ and $Cl^-$ ions are minor components of the total $PM_3$, and thus the AMS-PILS difference has minor influence on the calculations here.) Since the PILS was not functioning on 18[th] January, 2013, the ionic compositions of $K^+$ and $Cl^-$ were estimated from the linear relationship between PILS and AMS composition (Equations C1-C2). Since $Na^+$ measured by PILS was generally constant during the episode it was assumed to be the same on 18[th]. The diurnally varying concentrations

of total $NH_{3(g+p)}$ for ISORROPIA were calculated as the sum of $NH_4^+{}_{(p)}$ measured by AMS and $NH_{3(g)}$ measured by the denuder at the surface in Fresno; since the denuder measurements were averages over 6-7 hours, the $NH_{3(g)}$ concentration was linearly interpolated between the individual measurements to allow for estimation with higher time resolution. The 6-7 hour average denuder-based $NH_{3(g)}$ measurements compare reasonably with the point $NH_{3(g)}$ measurements made on board the P3-b at the

lowest altitude over Fresno. The fraction of $NO_3^-$ predicted to be in the gas-phase was also found to be in generally good agreement with the observations (Figure D1; (Parworth et al., 2017).

$$[Cl^-] = 1.24 * [Cl^-]_{AMS} \qquad (D1)$$

$$[K^+] = -0.036 * [Org]_{AMS} \qquad (D2)$$

As the boundary layer rises, starting around 8 am, and air from the RL is mixed into the surface air,

the instantaneous $NO_3^-{}_{(p)}$ concentration at the surface ($[NO_3^-{}_{(p)}]_{surf}$) is calculated as:





$$\left[NO_{3\ (p)}^{-}\right]_{surf,t} = \left[NO_{3\ (p)}^{-}\right]_{surf,t-1} + \left\{\left[NO_{3\ (p)}^{-}\right]_{surf,t-1} - \left[NO_{3\ (p)}^{-}\right]_{RL,t}\right\} \times \left\{1 - \frac{w_e \cdot \Delta t}{BLH}\right\} \qquad \text{(D3)}$$

where $t$ and $t$-1 represent the current and previous time steps respectively, *BLH* is the boundary layer height (m AGL), $w_e$ is the entrainment velocity and $[NO_{3\ (p)}^{-}]_{RL,t}$ is the concentration of $NO_{3\ (p)}^{-}$ in the layer of air that is entrained. Between 8 am and (approximately) noon, the vertical $NO_{3\ (p)}^{-}$ profile

5 within the remaining RL (above the instantaneous BLH) is assumed to remain unchanged from the early-morning observed profile. The vertical $NO_{3\ (p)}^{-}$ profile is updated to that observed during the second flight once the BLH (from the CLASS model) reaches the ML height observed around noon.

The daytime photochemical production of $HNO_3$ was calculated from Reaction DR1 (Burkholder et al., 2015, http://jpldataeval.jpl.nasa.gov).

$NO_2 + OH \rightarrow HNO_3$; $k_{OH} = 2.8 \times 10^{-11}$ cm$^{-3}$ molecule$^{-1}$ s$^{-1}$            (DR1)

The OH concentration at a given time step was assumed to scale with the solar radiation flux (*SR*) as:

15 $[OH]_t = \left(\frac{SR_t}{SR_{max}}\right)[OH]_{max}$                          (D4)

where the maximum daytime OH concentration is assumed to be $[OH]_{max} = 1 \times 10^6$ molecules cm$^{-3}$, after (Pusede et al., 2016). The rate coefficient for condensation of $HNO_{3(g)}$ onto suspended particulates, $k_{cond}$, was calculated based on collision theory (Seinfeld and Pandis, 2006) as:

$k_{cond} = \sum_i \beta_i * 2 * D_{p,i} * D * 10^{-4} * dN_i$                  (D5)





where the summation is over particle size, $D_{p,i}$ is the mean particle diameter in the size bin $i$ (m), and $dN_i$ is the number concentration in the size bin $i$ (m$^{-3}$). The term $\beta_i$ is the size-dependent Fuchs correction in the continuum regime, given by:

$$\beta_i = 0.75 * \frac{1+Kn}{Kn^2+1.283*Kn+0.75} \qquad (D6)$$

where $Kn = \lambda/D_{p,i}$, and $\lambda$ is the gas mean free path (65 nm). The parameter $D$ is the diffusion coefficient of HNO$_3$ gas in air (cm$^2$ s$^{-1}$) given by (De Andrade et al., 1992):

$$D = 10^{-4.7773} * T^{1.366} \qquad (D7)$$

where $T$ is ambient temperature (K). The corresponding evaporation rate coefficient ($k_{evap}$) is determined as:

$$k_{evap} = \frac{K_{eqm}}{k_{cond}} \qquad (C8)$$

where $K_{eqm}$ is the instantaneous (temperature and RH-dependent) equilibrium partitioning coefficient for ammonium nitrate. The rate coefficient for loss of gas-phase HNO$_3$ or NO$_3^-{}_{(p)}$ from dry deposition,

20    $k_{dep}$ (s$^{-1}$) is:



$$k_{dep} = \frac{v_d}{BLH} \tag{C9}$$

where $v_d$ is the deposition velocity (cm s$^{-1}$) and *BLH* is the time-dependent boundary layer height. The HNO$_{3(g)}$ deposition velocity has been shown to vary linearly with wind speed (Ma and Daggupaty, 2000). Here, it was assumed that:

$$v_d = 1 + 9 * \left(\frac{ws - ws_{min}}{ws_{max} - ws_{min}}\right) \tag{C10}$$

where *ws* is the observed wind speed, and $ws_{min}$ and $ws_{max}$ are the minimum and maximum values observed. The bounds of Eqn. C10 (lower limit $v_d = 1$ cm s$^{-1}$ and upper limit 10 cm s$^{-1}$) were chosen to span previously observed ranges. While the accuracy of the empirical Eqn. C10 is not known, we note that use of a constant $v_d$ of 0.07 cm s$^{-1}$ does not change the box model output substantially (Figure S13). Of course, if the actual $v_d$ is lower than estimated here the influence of dry deposition on NO$_3^-$$_{(p)}$ concentrations would be decreased. The NO$_3^-$$_{(p)}$ deposition velocity was assumed to be 0.01 cm s$^{-1}$, consistent with much slower deposition of particles than soluble gases such as HNO$_3$. Dry deposition occurred both during the daytime and nighttime.

**Author Contributions**

GP and CDC wrote the paper, with contributions from all authors. CP, XZ, HK, AJB, LDZ, JBN, QZ and CDC made the measurements. GP, THB, ICF and CDC contributed to the modelling efforts. CDC and QZ designed the project.

**Author Contributions**

The authors declare that they have no conflict of interest.



## Acknowledgements

This work was funded by the California Air Resources Board (14-307) and the DISCOVER-AQ campaign was supported by NASA. ICF is supported in part by California Agricultural Experiment Station (Hatch project CA-D-LAW-2229-H). The authors thank the entire DISCOVER-AQ science team for their contributions to the DISCOVER-AQ dataset. Profs. Sally Pusede (Univ. Virginia) and Ron Cohen (UC Berkeley), and Prof. Mike Kleeman (UC Davis) are thanked for their particularly useful comments and discussion.

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



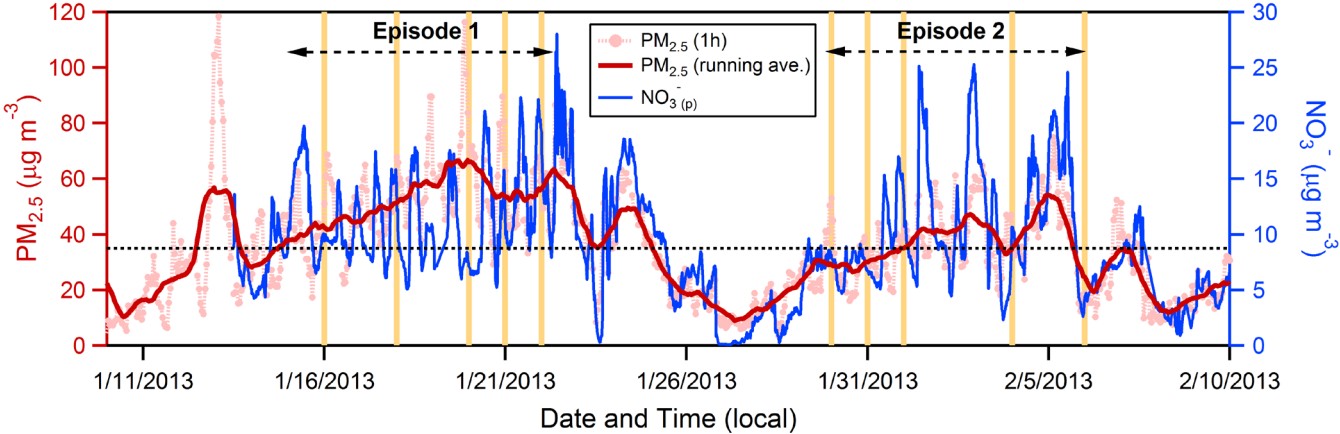

**Figure 1.** Time series of surface $PM_{2.5}$ concentration ($\mu g\ m^{-3}$) measured in Fresno during the DISCOVER-AQ campaign for 1 h averages (light red dotted line) and for a running average (red line; smoothed over 24 h), along with the 1 h average $NO_{3\ (p)}^{-}$ concentration (blue line). The vertical orange lines indicate the days on which airborne measurements were made. The horizontal dashed black line indicates the NAAQS 24 h standard of 35 $\mu g\ m^{-3}$ for $PM_{2.5}$.





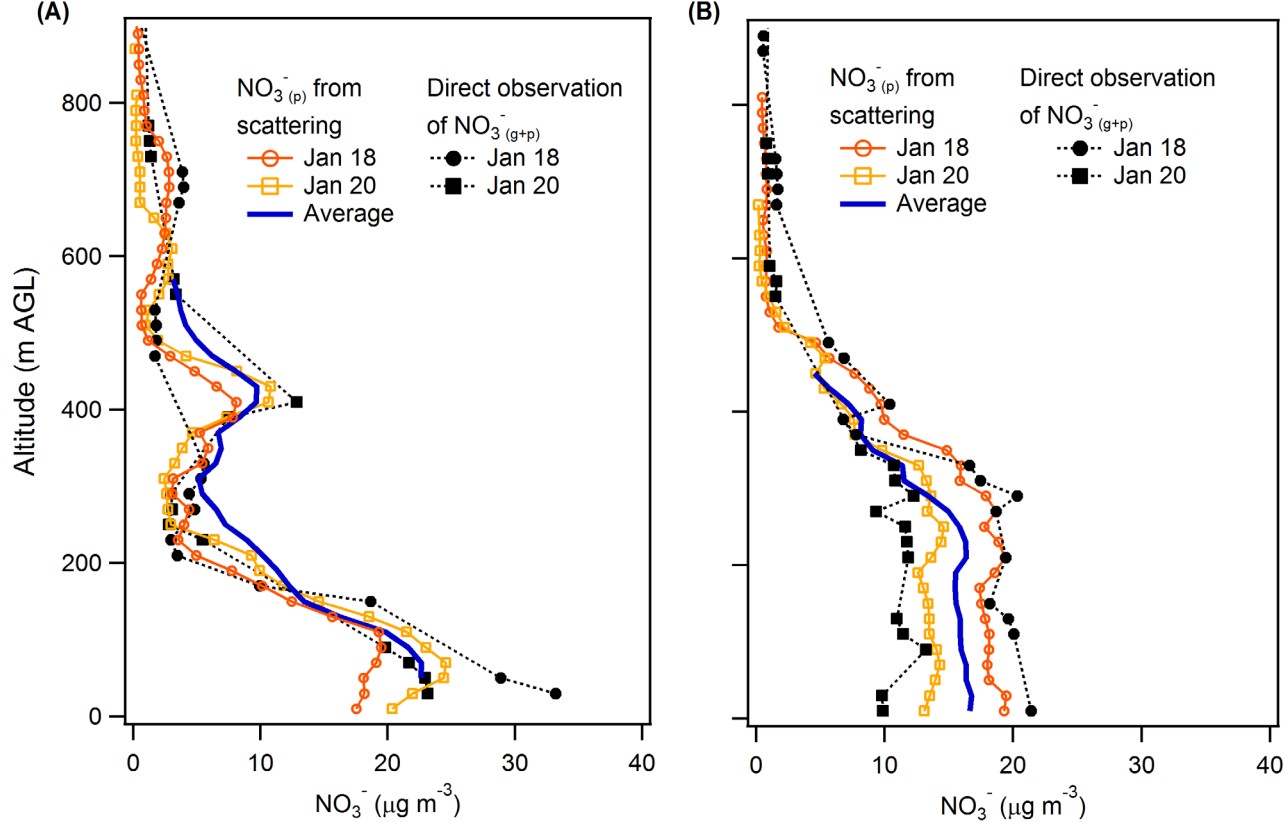

**Figure 2**. Vertical profiles for two individual flight days of particulate nitrate concentrations estimated from *in situ* total particle scattering measurements (open markers) and total nitrate (gas + particle) concentrations measured by the TD-LIF (solid black markers) for (A) the morning (~9:30 am) and (B) the afternoon ~2:30 pm. The solid blue lines indicate the average $NO_{3(p)}^-$ vertical profiles for all four flight days of Episode 1 (Jan 18, 20, 21 and 22).



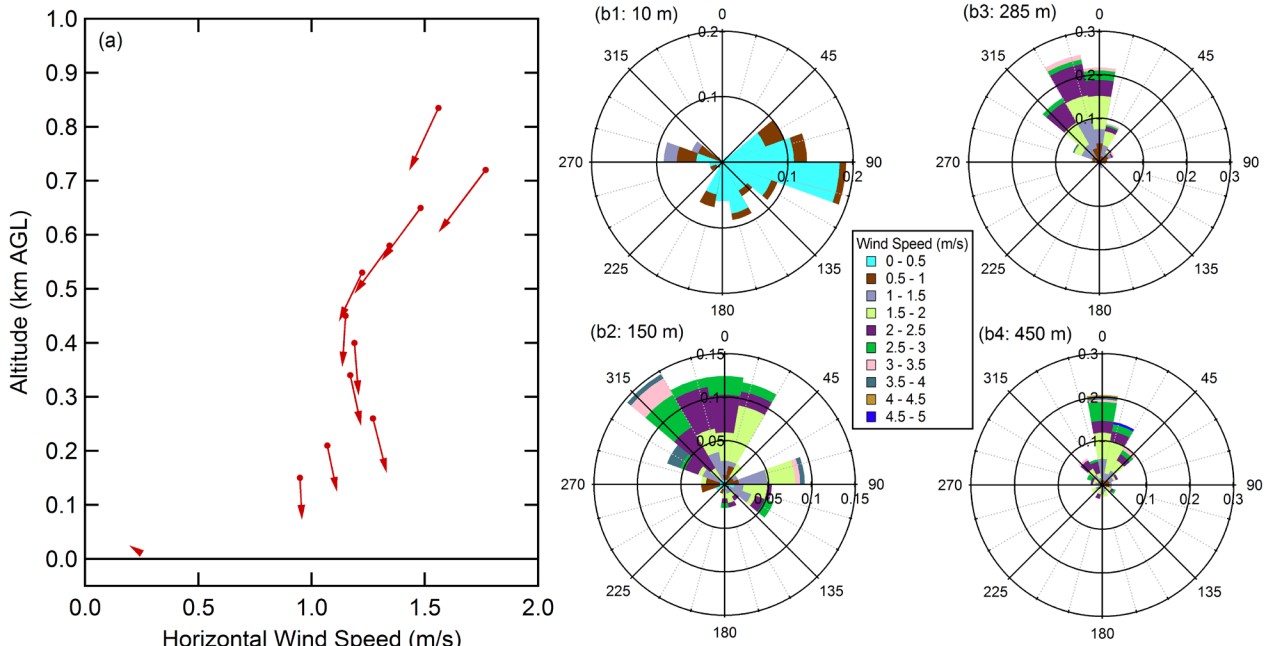

**Figure 3.** (a) Vertical profile of the average night time (19:00-07:00) horizontal winds over Visalia, CA (65 km SE of Fresno) and the surface (10 m) wind in Fresno for flight days during Episode 1 (Jan. 18, 20, 21, and 22). The length of the arrows corresponds to the wind speed and the direction to the average wind direction. (b) Corresponding wind roses for (b1) the surface, (b2) 125-175 m, (b3) 225-345 m, and (b4) 400-500 m. The length of each arc corresponds to the normalized probability and the colors indicate the wind speed (m/s; see legend). Data are from the National Oceanic and Atmospheric Administration, Earth System Research Laboratory, Physical Sciences Division Data and Image Archive (https://www.esrl.noaa.gov/psd/data/obs/datadisplay/, accessed 3 June 2017).





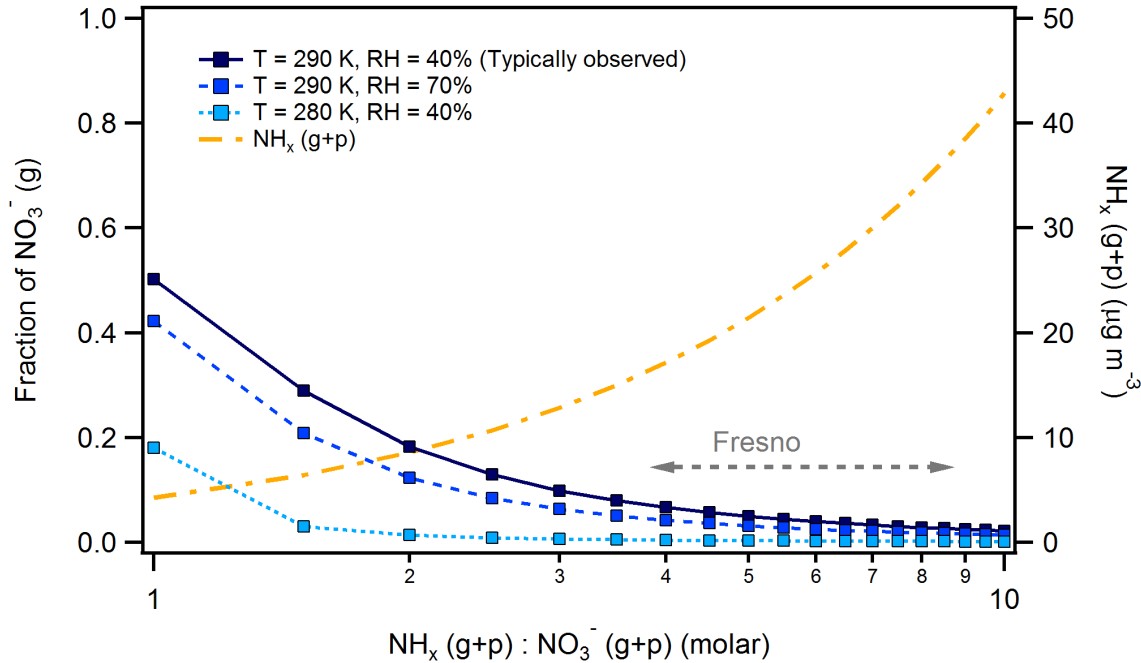

**Figure 4.** The gaseous fraction of total nitrate versus the molar ratio of total ammonia to total nitrate (ppb) under different environmental conditions (blue lines). The total ammonia is the sum of $NH_{3(g)}$ measured on P3-B close to ground (< 20 m AGL) and $NH_4^+{}_{(p)}$ at ground-level measured by PILS at approximately same time. The total nitrate is the $NO_3^-{}_{(g+p)}$ measured by TD-LIF close to ground (< 20 m AGL). The grey dashed arrow indicates the observed range of molar ratio values during the campaign period. The total (gas + particle) ammonia is shown for reference (orange line).



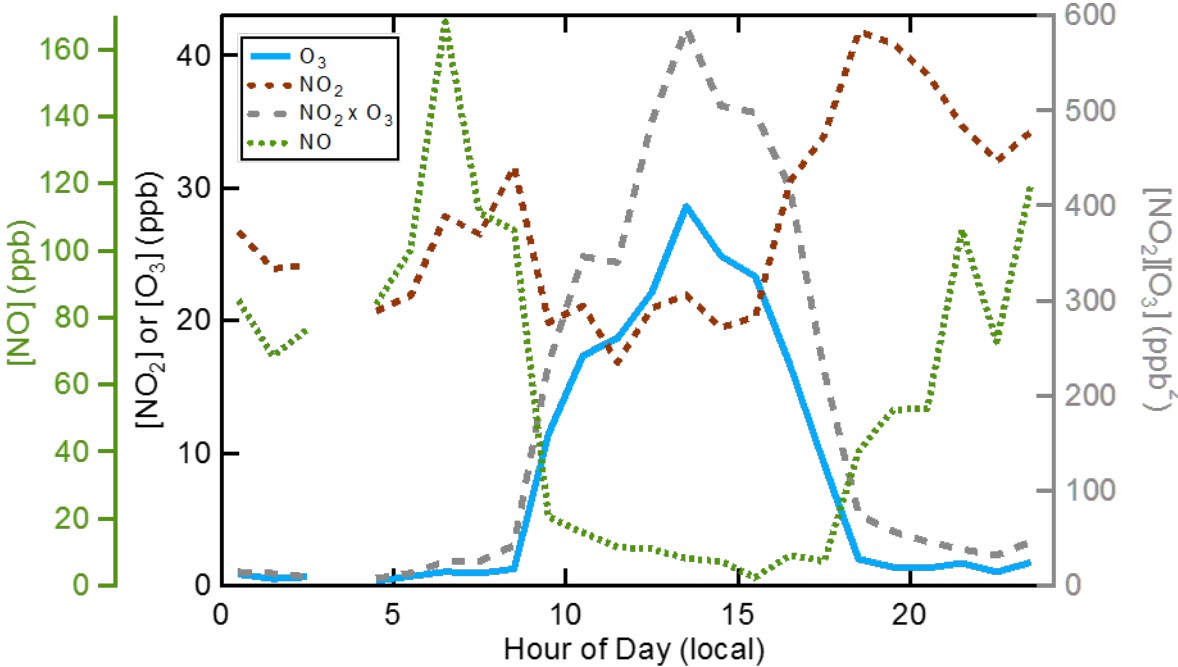

**Figure 5**. Diurnal profiles for ozone (blue), $NO_2$ (brown), NO (green) and the product of $O_3$ and $NO_2$ (gray) for the first pollution episode.



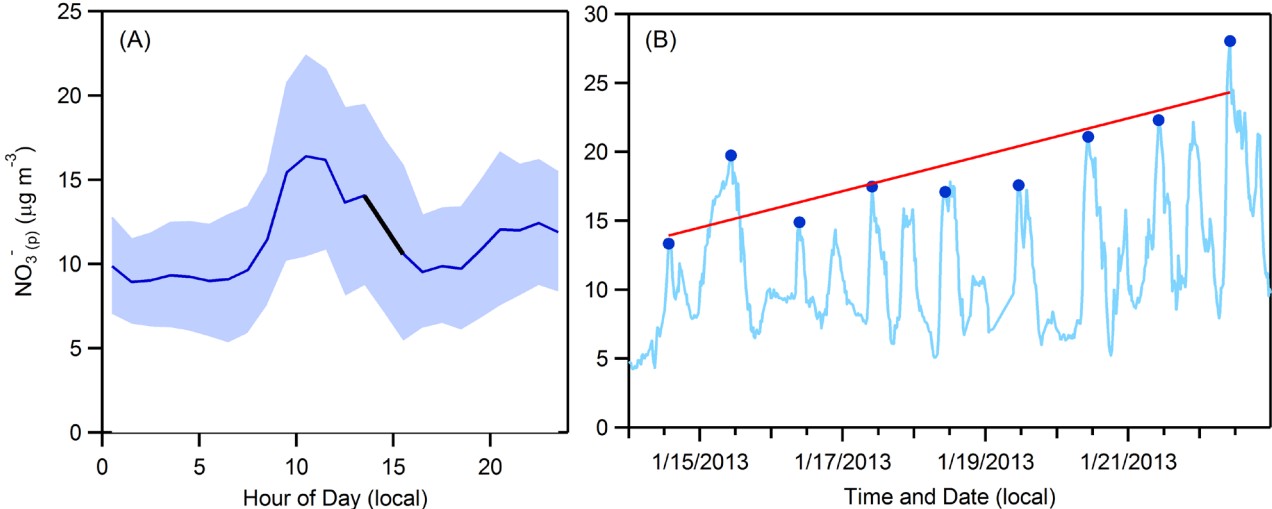

**Figure 6**. (A) Average diurnal profile (solid line) of surface $NO_3^-{}_{(p)}$ for all days of Episode 1. The shaded area indicates the $1\sigma$ standard deviation. The solid black line is a linear fit ($r^2 = 0.99$) to the data between 1:30 pm and 3:30 pm. (B) Time series (solid blue line) of surface-level $NO_3^-{}_{(p)}$ during Episode 1. The circles indicate the daytime peak values. The linear fit (red line) to the daytime $NO_3^-{}_{(p)}$ peaks suggest an increase of 1.32 µg m$^{-3}$ day$^{-1}$.





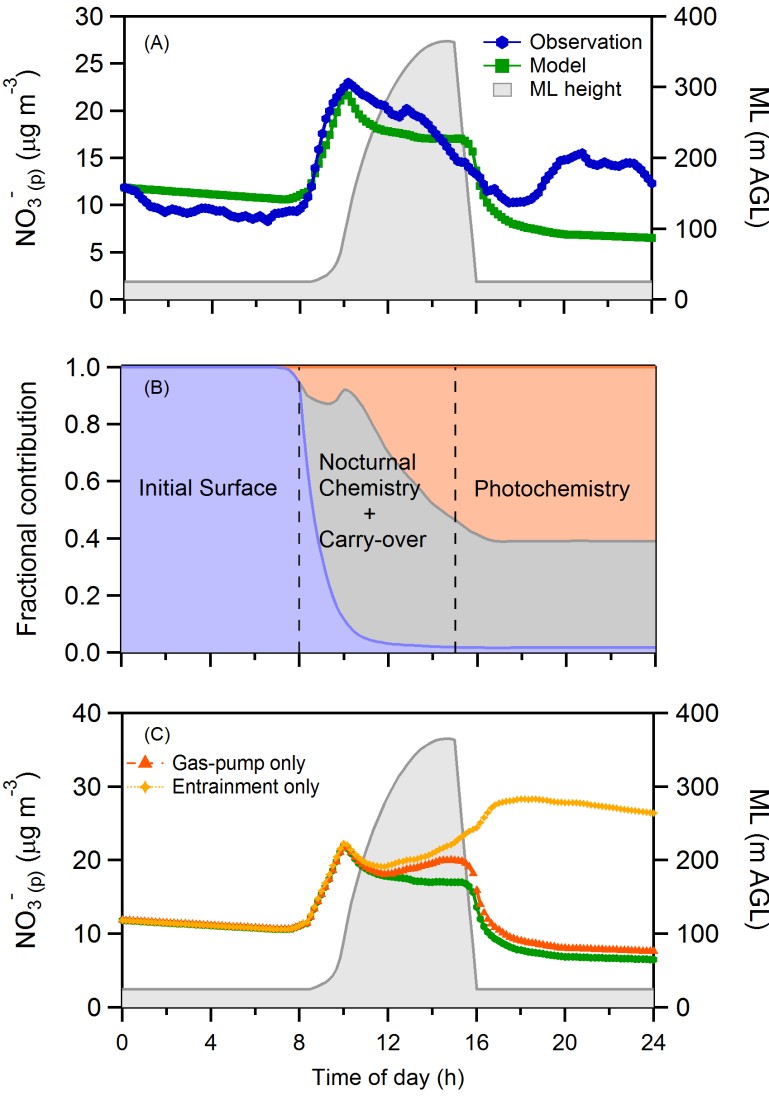

**Figure 7**. (A) Comparison between the observed (blue circles) and observationally constrained model predicted (green squares) diurnal profile of the surface $NO_3^-{}_{(p)}$ concentration ($\mu g\ m^{-3}$) for the four flight days (18[th], 20[th], 21[st] and 22[nd] January, 2013) during Episode 1. Also shown is the diurnal variation in
5   the boundary layer height (gray), as constrained by daytime measurements. (B) The diurnal variation in the simulated fraction of the total surface-level $NO_3^-{}_{(p)}$ contributed by the initial surface-level $NO_3^-{}_{(p)}$ (i.e. that at surface-level at 12:00 am), the $NO_3^-{}_{(p)}$ mixed down from the RL, and $NO_3^-{}_{(p)}$ produced from daytime photochemical reactions. (C) Comparison between the simulated diurnal profile when all processes are included (green squares, same as Panel A) and when only one $NO_3^-{}_{(p)}$ sink at a time is
10  considered. The individual sinks considered are only entrainment of free troposphere air (yellow crosses) or only dry deposition of $HNO_3$ via the gas-phase pump (orange triangles).





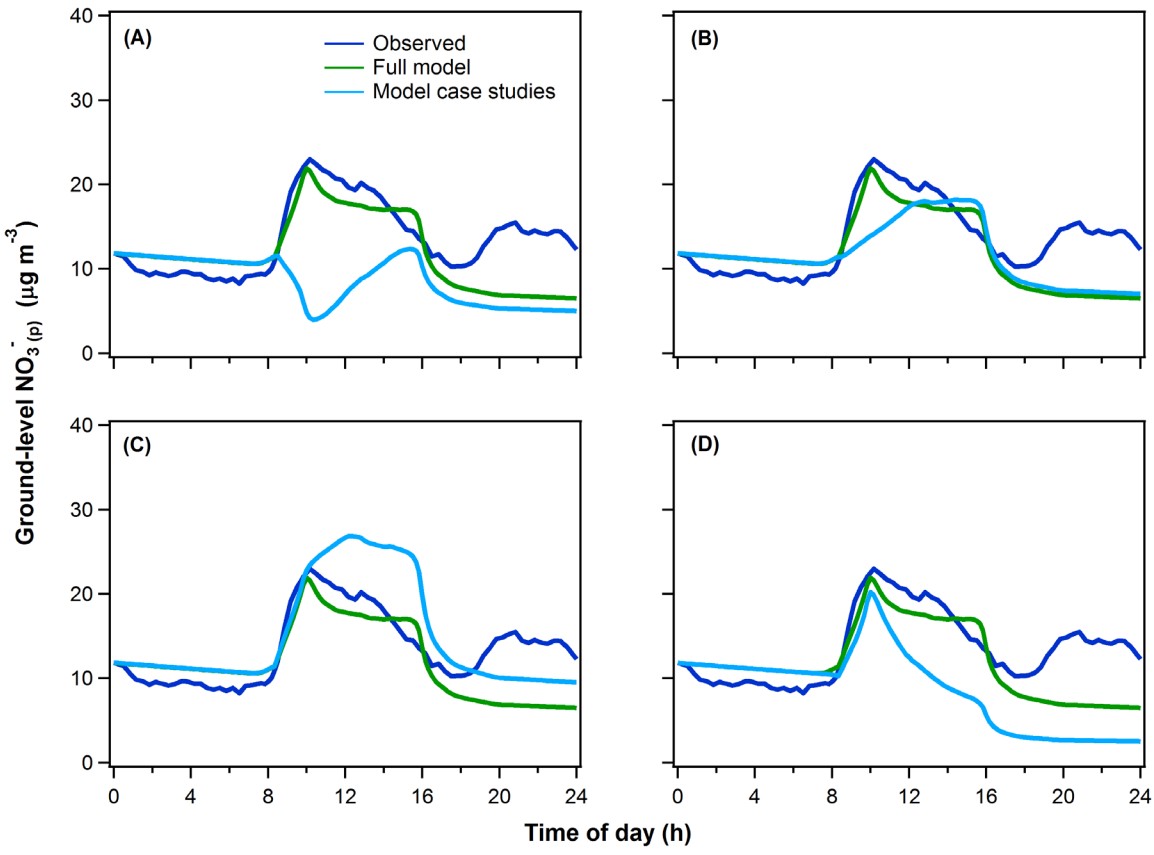

**Figure 8**. Model predictions of the diurnal variation in surface-level $NO_3^-{}_{(p)}$ under (A-C) different assumptions regarding the $NO_3^-{}_{(p)}$ concentration and vertical variability in the early-morning RL, or (D) without daytime photochemical production of $NO_3^-{}_{(p)}$. In all panels the blue curve shows the observations and the green curve shows the full observationally constrained model results (identical to Figure 6) for the average of the four flight days in Episode 1. For (A-C), the assumptions were: (A) The $[NO_3^-{}_{(p)}]_{RL}$ is equal to zero; (B) The $[NO_3^-{}_{(p)}]_{RL}$ is constant with altitude and equal to the $NO_3^-{}_{(g+p)}$ at 3 pm previous afternoon, corresponding to a case of zero net production or loss; (C) the $[NO_3^-{}_{(p)}]_{RL}$ is constant with altitude and equal to the maximum observed $[NO_3^-{}_{(p)}]$ in the early-morning RL profile.





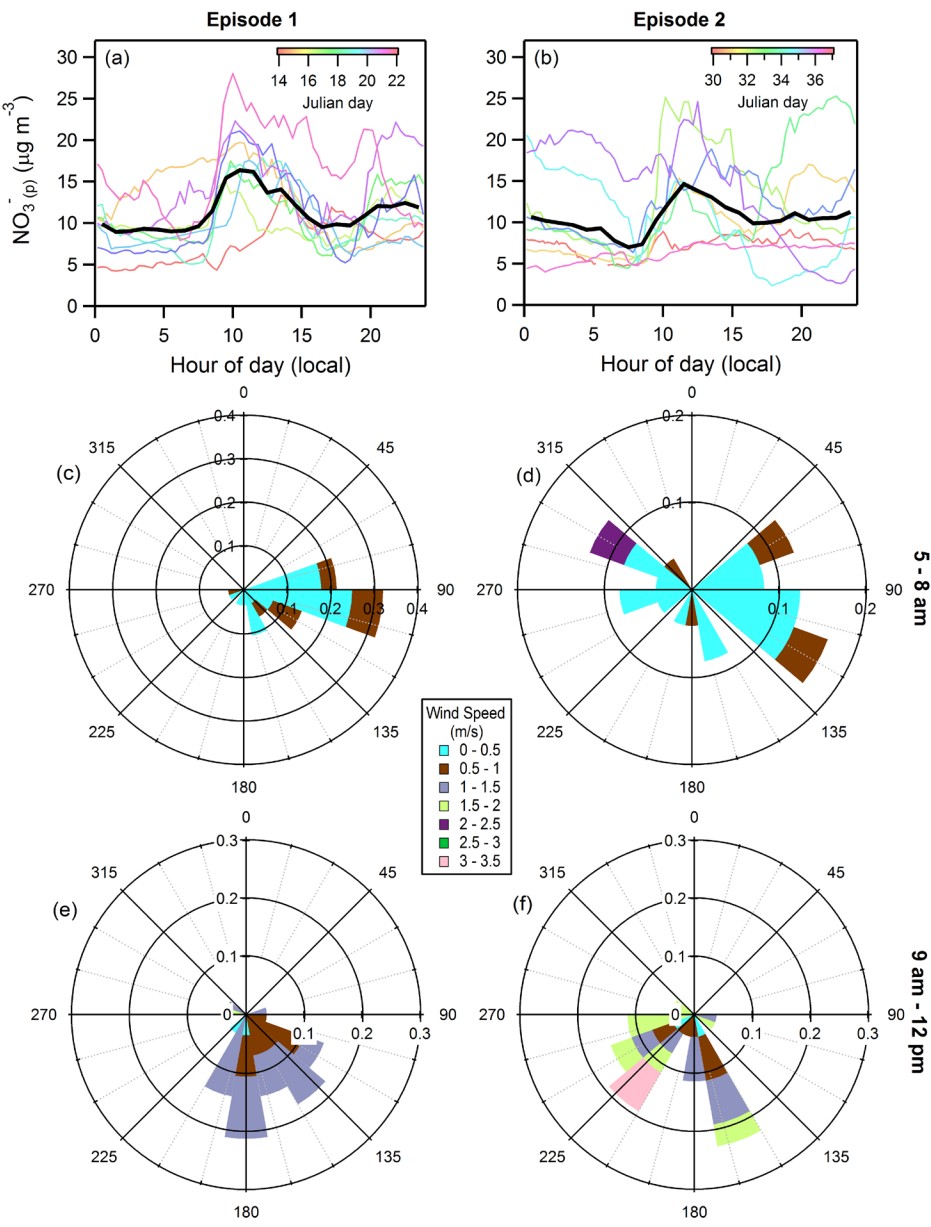

**Figure 9**. (Top panels) Diurnal variation in the surface-level particulate nitrate concentration during (a) the first episode and (b) the second episode. The solid black lines are the average profile over the episode and the colored lines are for individual days. (Middle/Bottom panels) Wind roses for surface-level (10 m) winds in Fresno for the early morning (5 – 8 am) during (c) episode 1 and (d) episode 2, and for the late morning (9 am – 12 pm) during (e) episode 1 and (f) episode 2.



**Table A1.** Summary of instruments deployed and measurements on ground and on aircraft made during the DISCOVER-AQ campaign.

| Platform | Measurement | Instrument | Uncertainty | Response time |
|---|---|---|---|---|
| NASA P3-B aircraft + Ground | Total and submicron scattering at 450, 550 and 700 nm | Integrating Nephelometer (TSI 3563) | 5% | 1 s |
| NASA P3-B Aircraft | Nitrate (gas+particle) | Thermal Dissociation - Laser Induced Fluorescence (TD-LIF) | 15% | 1 s |
| NASA P3-B Aircraft | Carbon monoxide (CO), Methane ($CH_4$) | Differential Absorption CO Measurement (DACOM) | < 2% | 1 s |
| NASA P3-B Aircraft | Nitrogen monoxide (NO), Nitrogen dioxide ($NO_2$), and Ozone ($O_3$) | 4-channel Chemiluminiscence | 10% for NO, 15% for $NO_2$, and 5% for $O_3$ | 1 to 3 s |
| NASA P3-B Aircraft | Ammonia ($NH_3$) | Picarro G2103 | 35% | 10 s |
| NASA P3-B Aircraft | Aerosol size distribution (0.06 – 1 μm) | Ultra-High Sensitivity Aerosol Spectrometer (UHSAS) | 20% | 1 s |
| NASA P3-B Aircraft | Meteorological and navigational measurements onboard | P3-B Project Data System (PDS) | - | 1 s |
| Ground | $PM_{2.5}$ mass concentration | Beta-Attenuation Mass (BAM) Monitor | 16% | 1 h |
| Ground | NO, $NO_2$ | Chemiluminiscence | 20% | 1 h |
| Ground | $O_3$ | NIST Standard Reference Photometer (SRP) | 2% | 1 h |
| Ground | Speciated non-refractory $PM_{1.0}$ | High Resolution Time-of-Flight Aerosol Mass Spectrometer (HR-ToF-AMS) | 25% | 5 min |
| Ground | Water-soluble components of $PM_{2.5}$ | Particle-Into-Liquid Sampler (PILS) coupled with two Ion chromatography systems | 10 - 20% | 20 min |
| Ground | Aerosol Particle Size | Scanning Mobility Particle Sizer (SMPS) | 10% | 1 min |
| Ground | Aerosol Particle Size | Aerodynamic Particle Sizer (APS) | 20% | 1 s |
| Ground | Refractive black carbon mass concentration | DMT Single Particle Soot Photometer (SP2) | 30% | 5 min |
| Ground | Relative humidity and temperature | | Temperature: ± 0.1 K RH: ± 2% | 1 h |



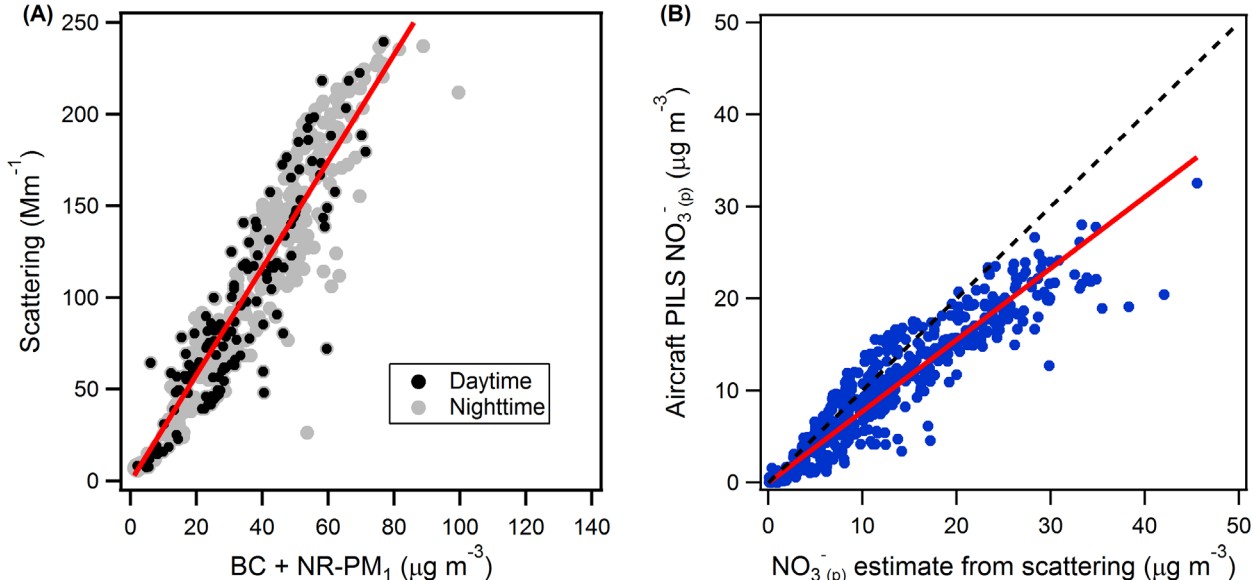

**Figure A1.** (A) Total particle scattering at 550 nm ($Mm^{-1}$) versus $PM_{1.0}$ mass (submicron black carbon, BC + non-refractory $PM_{1.0}$, $NR-PM_1$) concentration ($\mu g\ m^{-3}$) observed at ground-level in Fresno. The solid red line is the orthogonal distance regression fit including data only during the daytime (black circles) between 8 am and 4 pm; slope = 2.83 $Mm^2\mu g^{-1}$. (B) $NO_3^-{}_{(p)}$ concentration measured by PILS on P3-B aircraft versus that estimated from scattering using the relation $NO_3^-{}_{(p)} = \gamma \cdot \sigma_{sca,dry}/2.83$. The solid red line is the linear fit to the data, with slope = 0.78. The dashed black line is the 1:1 line.


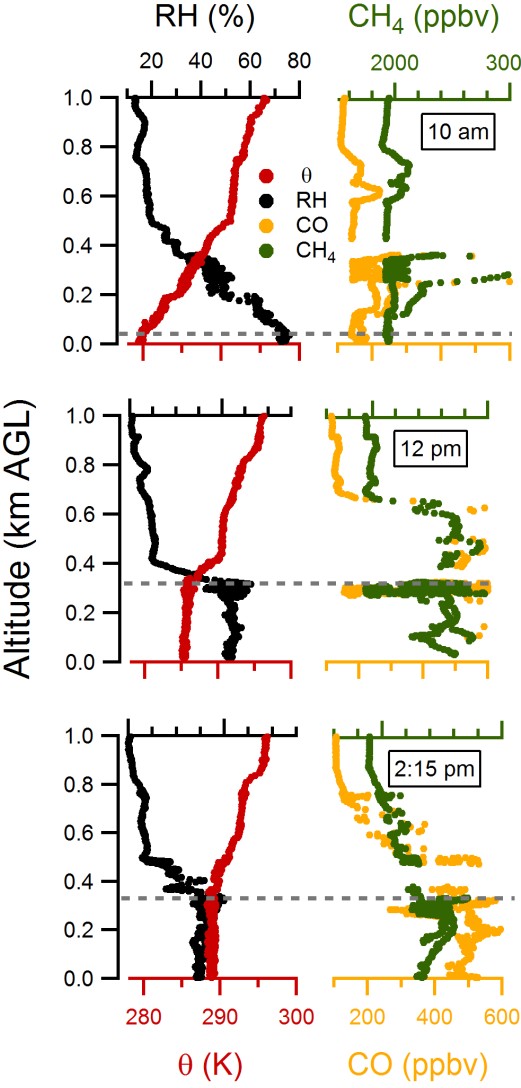

**Figure B1.** Vertical profiles of potential temperature, θ (K), relative humidity, RH (%), mixing ratios of carbon monoxide, CO (ppbv), and methane, CH₄ (ppbv) measured from the P3-B aircraft over Fresno on 18$^{th}$ January, 2013. The horizontal dashed grey line indicates the mixed boundary layer heights.





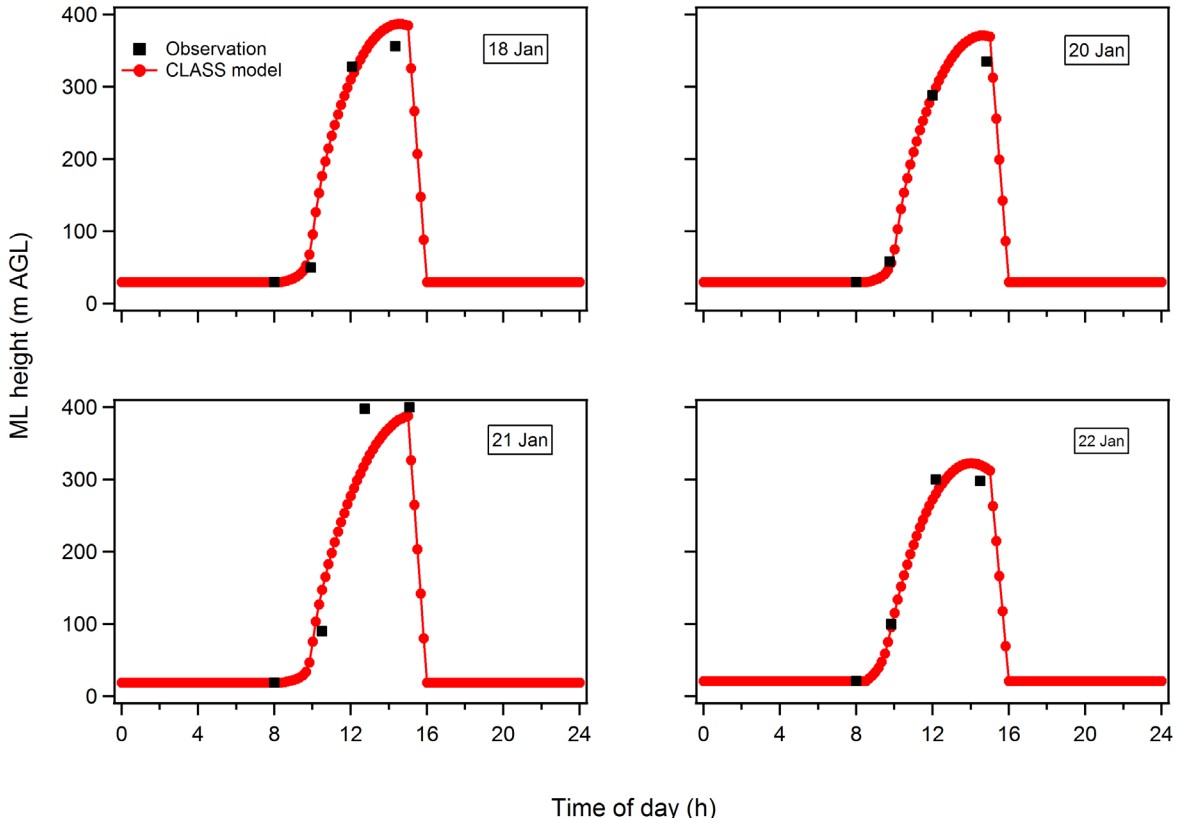

**Figure B2.** Evolution of the ML height with time (starting at 8 am) on the four flight days in Episode 1. The observational constraints are shown as black circles, where the first point comes from nearby balloon sonde measurements and the last three from the P3-B vertical profiles.





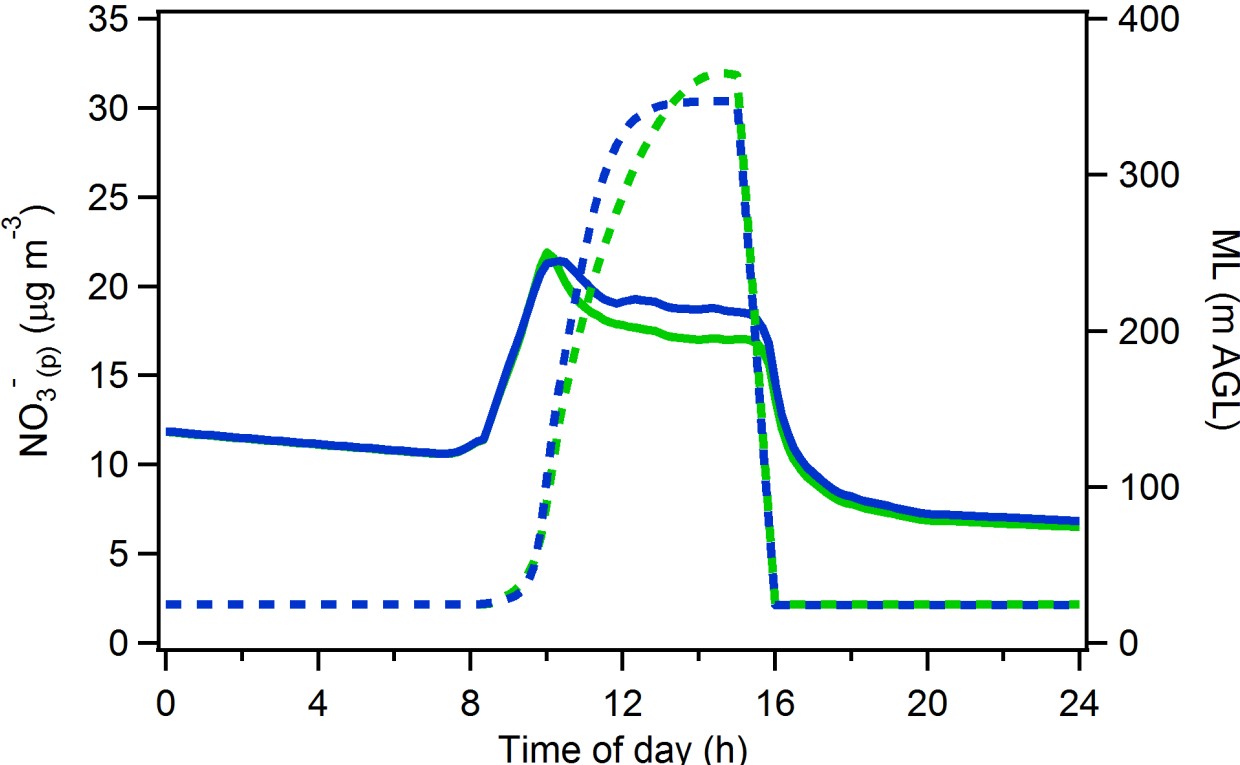

**Figure B3.** Average modelled surface $NO_3^-{}_{(p)}$ (solid lines) using the CLASS model output (green) and a sigmoid fit to the observed ML heights (blue). The ML heights used in the model are shown in dashed lines.



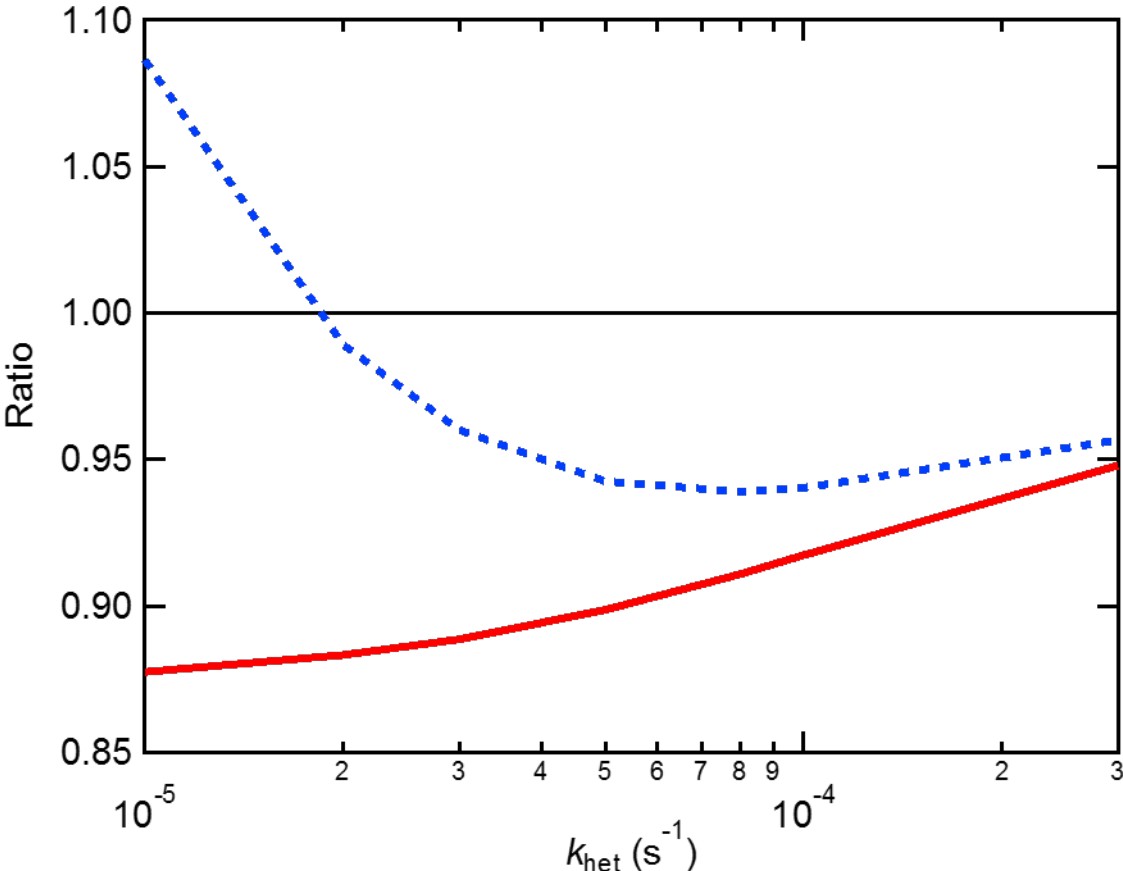

**Figure C1.** Model results showing the influence of including $NO_3$ + VOC reactions on $HNO_3$ production via the heterogeneous hydrolysis of $N_2O_5$, as a function of the heterogeneous oxidation rate. The red line shows the ratio between the $HNO_3$ produced via $N_2O_5$ hydrolysis when reactions with VOCs are considered and when they are not. Reaction of $NO_3$ with VOCs reduces the $HNO_3$ formed via hydrolysis. The blue line shows the ratio between the total $HNO_3$ produced from either $N_2O_5$ hydrolysis or $NO_3$ + VOC reactions when reactions with VOCs are considered and when they are not.



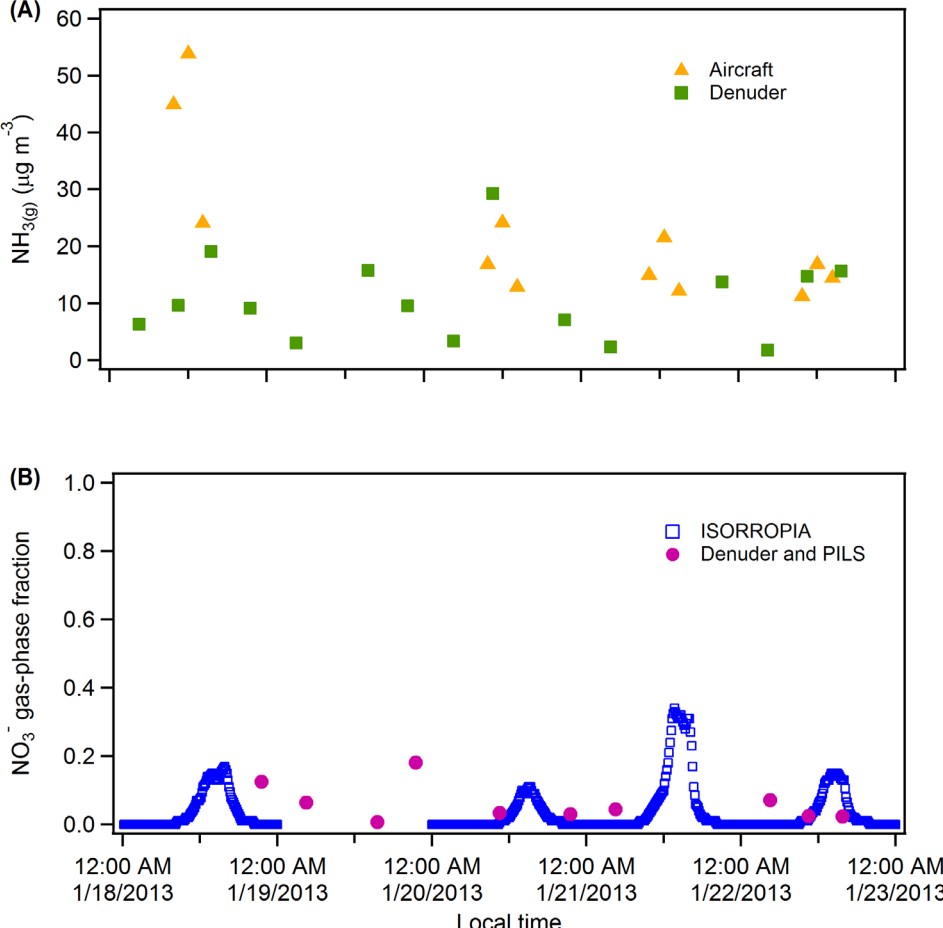

**Figure D1.** (A) Time series of $NH_{3(g)}$ ($\mu$g m$^{-3}$) measured with the denuder at the surface (green squares) and at the lowest altitudes by CIMS onboard P3-B aircraft (yellow triangles). (B) The nitrate gas-phase fraction estimated by ISORROPIA (blue squares) and the observed fraction determined from the denuder
5  $HNO_{3(g)}$ and PILS $NO_3^-{}_{(p)}$ measurements (pink circles) (Parworth et al., 2017).