# Peer review of "Observational assessment of the role of nocturnal residual-layer chemistry in determining daytime surface particulate nitrate concentrations"

_Atmospheric Chemistry and Physics, 2017_

## Referee Comment (RC1) · Anonymous Referee #1 · 21 Jul 2017

**Review of Observational assessment of the role of nocturnal residual-layer chemistry in determining daytime surface particulate nitrate concentrations by Prabhakar et al.**

The authors study the role of nocturnal chemistry and transport processes in determining surface PM2.5 levels and provide evidence for the morning transport of $NO_3^-(p)$ from the upper layers to the surface. This is a comprehensive study that includes an analysis of combined airborne and ground observations, and quantitative estimates of important parameters such as mixed layer height heights, loss processes of $NO_3^-$, and $N_2O_5$ uptake coefficients. I highly recommend the publication of this manuscript in ACP after minor revisions.

1. The authors suggest in Page 8 line 25 *".. the expected local nocturnal chemical production of nitrate in the RL should exhibit relatively minor vertical variation (due to variations in temperature and RH) (Figure S5). In other words, without loss or dilution processes, it is expected that the NO3-(p) concentration would increase to a similar extent at all RL altitudes."* and attribute the observed vertical distribution of $NO_3^-(p)$, which shows enhancements ~ 100 – 150 m agl at ~ 9:30 AM, to the differential horizontal advection in the RL. I am not 100% convinced that the horizontal differential advection is the main driver of the observed morning profiles of $NO_3^-(p)$. Figure 3a does NOT show maximum wind speed at ~ 250 m, which is also the altitude at which morning $NO_3^-(p)$ is minimum. Figure 3 suggests to me the strongest winds appear to be at 150 m very frequently, whereas on average they (b2 and b3) are basically the same through a deep layer and very light winds (~1.5 m/s). This is consistent with the idea that the horizontal transport aloft is responsible for the valley-wide distribution of pollutants/precursor gases, but it does not necessarily suggests its direct role in the vertical distribution of $NO_3^-(p)$. Also the average vertical profiles shown in Figure S3 and Figure 2A look very similar to the average nocturnal profiles of $NO_2$, $O_3$ and $N_2O_5$ reported by Brown et al. [1], indicating the altitude dependent chemical composition due to stable conditions could account for the observed $NO_3^-(p)$ profile if heterogeneous uptake of $N_2O_5$ is the key source of $NO_3^-(p)$. Consider re-phrasing some of the statements.

2. The authors assume morning chemistry due to ClNO2 and/or HONO is negligible even though the vertical profiling takes place 9:30 AM, after ~1 – 2 after sunrise and Brown et al. [2] reports enhancements in $NO_3^-(p)$ during the night and early morning. More justification or comment on morning chemistry is needed.
In the same vein, the authors derive [OH] by using peak [OH] of $1 \times 10^6$ molecules/cm$^3$ at noon, a relatively large number under wintertime conditions (although conditions in California might be different than other areas with typical winter season) and by scaling it with the observed solar radiation and estimate the photochemical contribution to be large, ~ 60% at 3 PM. Discussion on the effects of snow cover on the ground, cloud cover and fog during these episodes will be helpful since typical cold pool conditions often include those, which in turn limit the photochemical production.
In general, more extended discussion contrasting pollution episodes in other areas will improve the overall quality of this manuscript.

3. Transport out or entrainment from free troposphere in the afternoon leads to the loss of $PM_{2.5}$, but it also introduces $O_3$ and dilutes $NO_x$[3], impacting the $P_{NO3}$ formation at night.  What about the later effects?   Figure S3c indicates significant diffrences in O3 above and below the inversion cap during this episode.

4. Page 13 line 17: **"**The peak and turnover in surface-level NO3-(p) occurs when higher RL layers, where [NO3-(p)]RL < [NO3-(p)]ML, are entrained.**".**  Shouldn't it be "[NO3-(p)]RL >[NO3-(p)]ML"?

5. Using increase rate in 24-h $PM_{2.5}$ running average and % nitrate contribution might be more suitable for estimating the nocturnal nitrate production as it already take into consideration the transport term.

1.      Brown, S. S.; Dubé, W. P.; Osthoff, H. D.; Wolfe, D. E.; Angevine, W. M.; Ravishankara, A. R., High resolution vertical distributions of $NO_3$ and $N_2O_5$ through the nocturnal boundary layer. *Atmos. Chem. Phys.* **2007,** *7* (1), 139-149. DOI 10.5194/acp-7-139-2007.
2.      Brown, S. G.; Roberts, P. T.; McCarthy, M. C.; Lurmann, F. W.; Hyslop, N. P., Wintertime vertical variations in particulate matter (PM) and precursor concentrations in the San Joaquin Valley during the California Regional Coarse PM/Fine PM Air Quality Study. *J. Air Waste Manage. Assoc.* **2006,** *56* (9), 1267-1277. DOI 10.1080/10473289.2006.10464583.
3.      Baasandorj, M.; Hoch, S. W.; Bares, R.; Lin, J. C.; Brown, S. S.; Millet, D. B.; Martin, R.; Kelly, K.; Zarzana, K. J.; Whiteman, C. D.; Dube, W. P.; Tonnesen, G.; Jaramillo, I. C.; Sohl, J., Coupling between Chemical and Meteorological Processes under Persistent Cold-Air Pool Conditions: Evolution of Wintertime PM2.5 Pollution Events and N2O5 Observations in Utah's Salt Lake Valley. *Environ Sci Technol* **2017**, DOI 10.1021/acs.est.6b06603.

---

## Referee Comment (RC2) · Anonymous Referee #2 · 26 Jul 2017

This paper presents an observational analysis using flight and ground based obser-vations and 1-D box modeling to investigate nocturnal nitrate production above the boundary layer and the role it plays on the surface concentrations the next day. The paper relies primarily on aircraft data collected during one, multiday air pollution event in the San Jaoquin Vally, CA during wintertime. The focus is on understanding the impact of the chemistry and boundary layer processes, and the contribution of each on surface nitrate concentrations during winter, where approximately 30-80% of the PM mass is ammonium nitrate. Overall I find this to be an excellent paper and impactful to

many researchers that aim to investigate wintertime nitrate concentrations and want to understand how to regulate emissions to reduce the PM concentrations.

General Comments:

1) It would be helpful to have a better discussion related to how this paper fits into previous studies from Idaho, Washington, and Utah. The authors state in the abstract (page 2, lines 17-19) that the results from their paper "provide general insights into the evolution of pollution episodes in wintertime environments". However, the discussion that relates their findings to previous research in other areas is limited to two sentences (page 20, lines 1-5).

2) The surface heat flux and friction velocity are mentioned in the manuscript and appendix related to the calculation of mixing layer height, it is not clear what instrument this data come from, or if they came from a model. (page 12 line 10, page 25, lines 16-19)

3) The dataset might be too limited, but can the overnight advection be quantified and related to local sources? This would benefit the discussion on page 9 (lines 5-8) and the discussion about the advection from a nearby source on page 17 (lines 9-11).

Specific Comments:

1) Page 6, lines 15-16: "differential horizontal transport in the RL" sounds a bit awkward; the term advection might be suitable here.

2) Page 7, lines 22-24: "The derived, observationally constrained. . .." Referring to the estimated versus observed profiles of nitrate. While in general there is good agreement, there is not good agreement in the morning profiles close to the surface (<75m), especially in regard to the shape of the profiles. It would be worth mentioning that here.

3) Figure 4 is referenced in the text before Figure 3, these two figures should be switched.

[Figure]

4) Page 9, line 17-18: The maximum wind speed occurring at 250m is not clear based on the data shown in Figure 3a.

5) Page 12, lines 24-26: "good agreement" while the timing of the morning peak between the model and observations is captured the model does not capture increases in the surface nitrate concentrations after the ML decreases. While this is discussed later in the paper, it should be mentioned here that the model does not have good agreement with observations during the evening.

6) Page 17, line 9: Should this be referencing Figure S6 instead of Figure 8?

7) Page 19, lines 18-19: "vertical mixing has a particularly large impact on the ..." Is this vertical mixing really entrainment and dry deposition? The paper does not quantify or discuss modeling results for vertical mixing throughout the boundary layer; the box model focus is dry deposition and entrainment.

8) The sampling times for the instruments might be beneficial to the reader (Table A1), for example "Fast measurements. . ." (page 7, line 16) does not really have a context.

9) Page 25, lines 25-27: Is the assumption for the boundary layer to linearly drop over a 1-hour period reasonable, it seems too quick, and how does this assumption impact the results? This is vaguely referred to on page 17 (lines 19-22) where the decoupling in the model occurs very rapidly while the temperature and RH changes from observations appear to be more gradual. Is this, or could it potentially, have an impact on the evening increase in nitrate concentrations?

Technical Corrections:

1) There are a few (∼20-30) minor typos or grammatical errors.

---

## Author Response (AR1)

COLLEGE OF ENGINEERING

DEPARTMENT OF CIVIL & ENVIRONMENTAL ENGINEERING
ONE SHIELDS AVENUE
DAVIS, CALIFORNIA 95616
PHONE (530) 752-8180
FAX (530) 752-7872

21 September 2017

Dear Dr. Brown,

In response to the very helpful reviewer comments, we have made extensive changes to the manuscript. We believe that we have addressed all of the reviewers concerns.

We provide below our: (i) point-by-point response, (ii) the revised manuscript with changes tracked and (iii) the revised supplemental material with changes tracked.

Please let me know if you have any questions.

Regards,

Christopher D. Cappa
Professor and Vice Chair
Civil and Environmental Engineering
University of California, Davis

**Response to Reviewer Comments:**

We thank the reviewer for the suggestions on how to improve our paper. Our responses follow below, with reviewer comments in **black** and our responses in **blue**. Alterations to the manuscript text are in quotes with new additions in *italics*.

**Reviewer #1**

1. The authors suggest in Page 8 line 25 ".. the expected local nocturnal chemical production of nitrate in the RL should exhibit relatively minor vertical variation (due to variations in temperature and RH) (Figure S5). In other words, without loss or dilution processes, it is expected that the NO3-(p) concentration would increase to a similar extent at all RL altitudes." and attribute the observed vertical distribution of NO3-(p), which shows enhancements ~ 100 – 150 m agl at ~ 9:30 AM, to the differential horizontal advection in the RL. I am not 100% convinced that the horizontal differential advection is the main driver of the observed morning profiles of NO3-(p). Figure 3a does NOT show maximum wind speed at ~ 250 m, which is also the altitude at which morning NO3-(p) is minimum. Figure 3 suggests to me the strongest winds appear to be at 150 m very frequently, whereas on average they (b2 and b3) are basically the same through a deep layer and very light winds (~1.5 m/s). This is consistent with the idea that the horizontal transport aloft is responsible for the valley-wide distribution of pollutants/precursor gases, but it does not necessarily suggests its direct role in the vertical distribution of NO3-(p). Also the average vertical profiles shown in Figure S3 and Figure 2A look very similar to the average nocturnal profiles of NO2, O3 and N2O5 reported by Brown et al. 1, indicating the altitude dependent chemical composition due to stable conditions could account for the observed NO3-(p) profile if heterogeneous uptake of N2O5 is the key source of NO3-(p). Consider re-phrasing some of the statements.

While we appreciate the reviewer's concern, we disagree with at least the first part of the reviewer's argument. Figure 3A (now Figure 4A) very clearly shows a maximum in horizontal wind speed at 250 m. This is apparent in the Figure, redrawn on the next page to explicitly show lines for 150 m and 250 m. The mean horizontal wind speed at 250 m is clearly larger than at 150 m. We suspect there was some confusion as to what altitude is associated with each arrow. The altitude is indicated by the little circles at the start of the arrow, not the arrow head. Given that both reviewers had some difficulty with this figure, we have modified the caption to explicitly indicate that the little circles are the altitude, not the arrow heads: "The length of the arrows corresponds to the wind speed and the direction to the average wind direction, *with the measurement height indicated by the small circle on the tail of the arrow*." Our hope is that this change in the caption clarifies the altitude at which each measurement was made.

[Figure]

**Figure 1.** Annotated reproduction of Figure 3A from the original manuscript, showing the maximum horizontal wind speed is observed at 250 m.

Regarding the Brown et al. (2007) measurements, it should first be made clear that those measurements were made in the fall in a rural area of Colorado, not in wintertime CA over a medium-sized city such as Fresno. Regardless, they observed notable vertical structure in $NO_3$ and $N_2O_5$, indicative of "distinct chemical regimes as a function of altitude." One potentially important difference is that, unlike in Fresno, they observed substantial differences in the relative humidity with altitude. This could contribute to some of the vertical variation, although not likely the highly resolved structure that was occasionally seen by them in their series of overnight profiles.

To relate this to the Fresno measurements here, we emphasize that the chemical regime is not fundamentally separable from advection. This is especially so since both $O_3$ and $NO_2$ will start in the residual layer with approximately altitude-independent concentrations. If advection serves to bring in cleaner air (with respect to $NO_x$ and $O_3$) from outside the city then the $N_2O_5$ production would be reduced. Conversely, if $O_3$ and $NO_2$ are higher outside the city then $N_2O_5$ production might actually be enhanced. Brown et al. (2007) does state that "periodic advection of air masses containing large amounts of NO from nearby sources would be consistent with…, and is likely to be a large contribution to the observed gradients." Further, the role of advection is implicit in serving to create vertical variability in concentrations. We also note that the surface and 50 m nocturnal wind speeds reported by Brown et al. tended to be considerably higher than the wind speeds observed during DISCOVER-AQ. This could potentially explain some of the greater variability observed by them, especially since higher wind speeds would also tend to produce comparably higher nocturnal boundary layers, consistent with their

observations of potential temperature. For Fresno, $NO_2$ is clearly lower outside the city region (Pusede et al., 2014; Pusede et al., 2016). Measurements of $O_3$ and $NO_2$ made over the DISCOVER-AQ period are available for Madera (located NW of Fresno), Fresno and Parlier (located SE of Fresno) from the California Air Resources Board. (These cities were selected for comparison, based on the comprehensive summertime analysis of Pusede et al. (2014).) $O_3$ concentrations are slightly higher in the more rural cities around Fresno (Figure 2; added to the supplemental as Figure S6). The particle nitrate production efficiency scales approximately as $[NO_2][O_3]$ (all other factors being equal). This product is substantially higher in Fresno than in Madera and Parlier in the late afternoon, when boundary layer decoupling occurs (Figure 2; added to the supplemental as Figure S6). This suggests that advection from surrounding areas within the residual layer is likely to decrease the overall overnight particulate nitrate production over Fresno, as experienced the next morning when the residual layer air is entrained to the surface. However, we note that this assumes otherwise equivalent conditions (in particular, the loss rate of $N_2O_5$ with respect to uptake on particles). We also point to Figure S6 (now S7), which explicitly shows an inverse relationship between the nitrate concentration in the early-morning vertical profile and the overnight average wind speed at that altitude.

The above is all to say that we believe it is fully consistent with the observations of Brown et al. (2007) to make an argument that advection is a driving force in shaping the overnight the local vertical profile, since here the calculations of nitrate production based on observationally constrained initial conditions indicate that chemical or physical (T & RH) differences alone are unlikely to explain the observations. Thus, advection can have a secondary impact of changing the chemical conditions in addition to the direct impact of importing air with a lower particulate nitrate concentration. This was originally discussed on P8,L25 through P9,L11. We have revised this section to try and make these points more clearly, as:

> "Box model calculations indicate that the expected local nocturnal chemical production of nitrate in the RL should exhibit relatively minor vertical variation due to variations in temperature and RH *alone* (Figure S5). In other words, without *advective* loss or dilution processes *of either $NO_{3^-(p)}$ or the precursor gases* it is expected that the $NO_{3^-(p)}$ concentration would increase to a similar extent at all RL altitudes.

> The substantial changes observed in the shape of the vertical profile overnight indicate that night time differential advection in the RL is a major factor in determining the shape of the morning $NO_{3^-(p)}$ vertical profile during this pollution episode. Differential horizontal advection serves to *directly* export $NO_{3^-(p)}$ from the urban area and import cleaner air from surrounding areas. *Secondarily, as $NO_x$ concentrations are also lower outside of the Fresno urban area (Pusede et al., 2014),* this differential advection will also influence the over-city concentrations of precursors gases ($NO_x$ and $O_3$; Figure S3-S4) and consequently the *altitude-specific* nitrate production, with decreases likely. *This is supported by surface-level measurements of*

*NOx and O3 made in Fresno and in the nearby and much more rural cities of Parlier (located 35 km SE of Fresno) and Madera (located 40 km NW of Fresno). The NOx and NO2 concentrations are higher and the O3 lower in Fresno compared to the surrounding cities throughout the day, and the instantaneous nitrate production rate ([NO2][O3]) is substantially higher in Fresno in the late afternoon, when decoupling occurs (Figure S6).* The important implication is that overnight advection *both directly and indirectly alters the vertical NO3⁻(p) profile and* decreases the over-city NO3⁻(p) concentrations in the morning, which will consequently serve to limit the extent of localized pollution build-up during events."

[Figure]

Figure 2. Diurnal profiles in $NO_2$, $O_3$, NO and $NO_2$ x $O_3$ concentrations for Fresno, Parlier and Madera.

2a. The authors assume morning chemistry due to ClNO2 and/or HONO is negligible even though the vertical profiling takes place 9:30 AM, after ~1 – 2 after sunrise and Brown et al. 2 reports enhancements in NO3-(p) during the night and early morning. More justification or comment on morning chemistry is needed.

As stated by (Brown et al., 2006), "Surface aerosol nitrate data were not of sufficient quality and are not used." Thus, they only report on measurements of particulate nitrate made at 90 m AGL, not at the surface. This is an important distinction since we find that the nocturnal boundary layer height was << 90 m AGL for the DISCOVER-AQ measurements and we are reporting on diurnal variation in surface concentrations. This key difference makes it difficult to compare the Brown et al. (2006) observations with our observations. Their comparison between $O_3$ and NO concentrations measured at 90 m and 7 m AGL certainly suggests that they were sampling particles above the surface layer when sampling at 90 m. As such, we would fully expect to observe different diurnal behavior between their study and ours. Their observing of overnight "enhancements" in particulate nitrate are fully consistent with the mechanism that we are proposing. We also note that Brown et al. (2006) do not report diurnal profiles for their particulate nitrate measurements, despite reporting them for many other parameters (e.g. particle number, black carbon, $O_3$, NO). It is thus quite difficult to discern any specific timing with respect to time of day from their measurements, although there does seem to be a midmorning peak on many days in their time series.

Regarding HONO, (also relevant to 2b below), we note that the estimated peak OH includes contributions of HONO photolysis, which can actually dominate the OH production (Kim et al., 2014b). We now indicate this specifically as:

> "Photochemical production of HNO3 is calculated based on the oxidation of NO2 by hydroxyl radicals, with wintertime concentrations estimated to peak around [OH] = $10^6$ molecules cm 3 at noon in the region, *with contributions from O($^1$D) + H$_2$O (from O$_3$ photolysis), HONO photolysis and CH$_2$O photolysis* (Pusede et al., 2016)."

2b. In the same vein, the authors derive [OH] by using peak [OH] of 1x106 molecules/cm3 at noon, a relatively large number under wintertime conditions (although conditions in California might be different than other areas with typical winter season) and by scaling it with the observed solar radiation and estimate the photochemical contribution to be large, ~ 60% at 3 PM. Discussion on the effects of snow cover on the ground, cloud cover and fog during these episodes will be helpful since typical cold pool conditions often include those, which in turn limit the photochemical production.

The assumption of [OH] = 1e6 molecules cm$^{-3}$ was based on the more detailed chemical calculations of Pusede et al. (2016), and includes contributions from O($^1$D) + H$_2$O (from O$_3$ photolysis), HONO photolysis and CH2O photolysis. For broader context, we are aware of three observational studies reporting midday [OH], with: 1.4e6 molecules cm$^{-3}$ in New York City (Ren et al., 2006), and 1.7e6 molecules cm$^{-3}$ in Birmingham, UK (Heard et al., 2004), 2.7e6 molecules cm$^{-3}$ in Colorado (Kim et al., 2014a).

We note that there was no snow on the floor of the SJV during the campaign (and almost never is). There was little to no fog during DISCOVER-AQ, a contrast from some other years and a result of the regional climatology (e.g. el Nino vs. la Nina) (Young et al., 2016). Cloud cover was minimal. We have added the following to the "Materials and Methods" section of the manuscript: *"Local conditions during DISCOVER-AQ were relatively cool (Tavg = 7.9 °C) and dry (RHavg = 69%) with frequent sunshine and no visible fog."*

2c. In general, more extended discussion contrasting pollution episodes in other areas will improve the overall quality of this manuscript.

We have added a new section "3.4 Linking to other regions." Page 20, Lines 1-5 have been moved to this section and additional discussion is now provided. The new section is provided here:

**3.4 Linking to other regions**

[revised manuscript text omitted]

3. Transport out or entrainment from free troposphere in the afternoon leads to the loss of PM2.5, but it also introduces O3 and dilutes NOx, impacting the PNO3 formation at night. What about the later effects? Figure S3c indicates significant differences in O3 above and below the inversion cap during this episode.

Indeed, entrainment of FT air alters concentrations of $O_3$ and $NO_x$ in the mixed boundary layer, which then becomes the residual layer. However, as we use the observations of $O_3$ and $NO_2$ to constrain the formation model, and since the daytime boundary layer is well-mixed, the influence of daytime entrainment on at least the initial conditions in the residual layer is accounted for. To the extent that there is any entrainment of FT air into the residual layer at night this would impact chemical production in the residual layer. Given the strong atmospheric stability at night, nighttime entrainment of FT air into the residual layer is not expected to be

substantial, just as mixing between the residual layer and the nighttime boundary layer is strongly limited. (This point was already noted on P8 when we stated that "In the absence of a strong jet aloft and no convective mixing, night time entrainment of cleaner FT air into the RL is expected to be considerably slower than horizontal advection.") To make this point further, we have now added the following to the main text description of the box model: "*While entrainment of FT air also alters the NO2 and O3 concentrations in the mixed layer, since these are constrained by the surface observations (within the mixed layer) this is accounted for.*"

4. Page 13 line 17: "The peak and turnover in surface-level NO3-(p) occurs when higher RL layers, where [NO3-(p)]RL < [NO3-(p)]ML, are entrained.". Shouldn't it be "[NO3-(p)]RL >[NO3-(p)]ML"?

No. The increase up to the peak occurs due to entrainment of air with [NO3-(p)]RL >[NO3-(p)]ML. However, the actual peak occurs when air above the especially nitrate rich layer is entrained to the surface and where this higher-level air has [NO3-(p)]RL < [NO3-(p)]ML. The help clarify this, we now indicate that this "occurs when *even* higher RL layers" are entrained.

5. Using increase rate in 24-h PM2.5 running average and % nitrate contribution might be more suitable for estimating the nocturnal nitrate production as it already take into consideration the transport term.

We assume this comment is in reference to using the early-morning peak values (Figure 6B) to characterize the increase. Since we have direct measurements of particulate nitrate concentrations at the surface, it seems more straightforward to use those measurements, rather than PM2.5 and the nitrate contribution (since these are all related). As an alternative to the peak analysis, we have fit the 24-h average particulate nitrate concentrations over the episode. This gives a slope of 0.66 ug m$^{-3}$ d$^{-1}$, as opposed to 1.32 ug m$^{-3}$ d$^{-1}$ for the peak analysis. This is now stated on P18 as: "*For comparison, the 24-h average surface-level NO3 (p) increases by 0.66 µg m$^{-3}$ day$^{-1}$.*"

**Reviewer #2:**

We thank the reviewer for their helpful comments. Upon comparing the line numbers referenced by the reviewer with the line numbers in the ACPD paper, it appears that the reviewer may have incidentally been reading the initial submission that did not include changes in response to the Quick Review process. We note this simply to avoid confusion in terms of line numbers/figure numbers. We will refer to the published ACPD version in our response.

1. It would be helpful to have a better discussion related to how this paper fits into previous studies from Idaho, Washington, and Utah. The authors state in the abstract (page 2, lines 17-19) that the results from their paper "provide general insights into the evolution of pollution

episodes in wintertime environments". However, the discussion that relates their findings to previous research in other areas is limited to two sentences (page 20, lines 1-5).

We have added a new section "3.4 Linking to other regions." Page 20, Lines 1-5 have been moved to this section and additional discussion is now provided.

2. The surface heat flux and friction velocity are mentioned in the manuscript and appendix related to the calculation of mixing layer height, it is not clear what instrument this data come from, or if they came from a model. (page 12 line 10, page 25, lines 16-19)

We accidentally failed to state that the surface heat flux and friction velocity were both constrained by sonic anemometer measurements made at the Huron site, in addition to the radiosonde measurements. This information has now been added.

3. The dataset might be too limited, but can the overnight advection be quantified and related to local sources? This would benefit the discussion on page 9 (lines 5-8) and the discussion about the advection from a nearby source on page 17 (lines 9-11).

It is difficult to be much more quantitative regarding the particular role of advection from this dataset. The wind profiler data available from nearby Huron provide guidance, but as they are not co-located with the ground site it is difficult to be more explicitly quantitative. We do note here that we found that our back-calculated $N_2O_5$ uptake coefficient was lower than the expected uptake coefficient based on composition, which is consistent with advection having decreased local concentrations relative to the total potential nitrate formation overnight in the absence of advection. Regarding the discussion on P17, we have worked to understand the particular source of this evening "bump" in greater detail, for example using HYSPLIT back trajectories to identify likely source locations, but no clear answer has developed from this. One of the challenges here is that we do not know, for example, whether this late evening increase is likely the result of (for example) some source located close and transported a short distance over a short period of time or located further away and transported over a longer distance and time. Assessment is further complicated by the strong shift in wind direction between the surface (10 m) and higher altitudes (Figure 3 in the initial manuscript, now Figure 4). Since the focus of the manuscript is on the early morning and afternoon behavior, rather than this late evening phenomenon, we have limited the discussion regarding the nature or location of the nearby source.

4. Page 6, lines 15-16: "differential horizontal transport in the RL" sounds a bit awkward; the term advection might be suitable here.

We have changed this to "altitude-dependent advection" so as to continue to emphasize the idea that the influence of advection is not the same at all layers.

5. Page 7, lines 22-24: "The derived, observationally constrained..." Referring to the estimated versus observed profiles of nitrate. While in general there is good agreement, there is not good agreement in the morning profiles close to the surface (<75m), especially in regard to the shape of the profiles. It would be worth mentioning that here.

We have clarified the comparison to be more precise as follows:

"The derived, observationally constrained NO3 (p) profiles based on the estimated NO3 (p) exhibit generally good correspondence with the sparser direct measurements of NO3-(g+p), *although on one of the two days available for comparison the total $NO_3^-$ somewhat exceeds the estimated $NO_3^-(p)$ below ~75 m* (Figure 2)."

6. Figure 4 is referenced in the text before Figure 3, these two figures should be switched.

We have switched the order of these figures.

7. Page 9, line 17-18: The maximum wind speed occurring at 250m is not clear based on the data shown in Figure 3a.

We have added the word "local" to indicate that there is a local, not global, maximum at ~250 m height. Also, in Figure 1 above in this response we point out the local maximum. We have also modified the caption as follows: "The length of the arrows corresponds to the wind speed and the direction to the average wind direction, *with the measurement height indicated by the small circle on the tail of the arrow*."

8. Page 12, lines 24-26: "good agreement" while the timing of the morning peak between the model and observations is captured the model does not capture increases in the surface nitrate concentrations after the ML decreases. While this is discussed later in the paper, it should be mentioned here that the model does not have good agreement with observations during the evening.

We now state: "The model predictions for the individual flight days also exhibit generally good agreement with the NO3 (p) observations *except in the late evening, discussed further below* (Figure S7)."

9. Page 17, line 9: Should this be referencing Figure S6 instead of Figure 8?

This should have been Figure S7, not Figure 8. This has been corrected.

10. Page 19, lines 18-19: "vertical mixing has a particularly large impact on the …" Is this vertical mixing really entrainment and dry deposition? The paper does not quantify or discuss modeling results for vertical mixing throughout the boundary layer; the box model focus is dry deposition and entrainment.

We have clarified this statement in the revised manuscript. What was intended here is an indication that the very shallow nocturnal boundary layer means that vertical mixing in the morning, which entrains air from the residual layer, can strongly control the surface concentrations. We now state:

"vertical mixing *in the early morning, which entrains air from the residual layer into the surface mixed layer*, has a particularly large impact on the surface concentrations here due to the nocturnal boundary layer being exceptionally shallow."

11. The sampling times for the instruments might be beneficial to the reader (Table A1), for example "Fast measurements: : :" (page 7, line 16) does not really have a context.

We have revised this to say: "Fast measurements of total $NO_3^-$ (gas + particle, $NO_3^-$ (g+p)) were only available for a subset of flights (Pusede et al., 2016), and particulate-only NO3 measurements were not made with sufficient time resolution*, less than about a minute,* to allow for robust characterization of the $NO_3^-$(p) vertical profile."

12. Page 25, lines 25-27: Is the assumption for the boundary layer to linearly drop over a 1-hour period reasonable, it seems too quick, and how does this assumption impact the results? This is vaguely referred to on page 17 (lines 19-22) where the decoupling in the model occurs very rapidly while the temperature and RH changes from observations appear to be more gradual. Is this, or could it potentially, have an impact on the evening increase in nitrate concentrations?

The fall in the mixed layer height occurs much faster than the temperature and RH changes, as it is strongly related to the input of solar radiation, although the rapid fall in the mixed layer height is apparent in the rapid rise in $NO_x$ and fall in $O_3$ at this time. The decrease in the mixed layer height can occur very rapidly as the solar flux decreases since there is then a distinct lack of thermal forcing. Thus, despite the temperature remaining elevated for many hours after the sun goes down, there is not an energy input to sustain a high boundary layer height. This general phenomenon is evident in the classic figure from Stull (redrawn on Wikipedia, pasted below for reference). More specific to the SJV, Bianco et al. (2011) observed diurnal profiles of boundary layer heights across summer-winter seasons for one year using radio acoustic sounding system (RASS) profiles at various sites in the SJV. At the site located nearest to Fresno, Chowchilla, they observed that the daytime BLH was approximately constant during the winter, with perhaps a slight drop ~2 h before sunset followed by a rapid drop in the hour right around sunset. This behavior is generally consistent with our model framework. It is also consistent with turbulent kinetic energy decay rates in the late afternoon/evening transition as reported in

Lothon et al. (2014) and Nadeau et al. (2011), with TKE falling by an octave or more in an hour. Certainly the linear decline is an approximation to the true behavior. We have added the following to Appendix B to indicate this: *"A relatively rapid (~1 h) decline in the mixed layer height is consistent with wintertime observations of diurnal BLH profiles (Bianco et al., 2011)."*

In terms of the implications of this approximation/assumption, this would, however, not have an impact on the late evening increase in particulate nitrate. But, if we assumed a slower decline (such that the BLH remained higher for longer) then the calculated late afternoon decrease in surface concentrations of particulate nitrate due to the "gas-phase pump" and $HNO_3$ deposition would be decreased (c.f. equation C9, which has BLH in the denominator). Alternatively, if the BLH fell more rapidly, then the calculated late afternoon decrease in particulate nitrate would have been greater. We aimed to make this point in the original manuscript with the parenthetical that stated (P17,L4-7):

"(In the model here, the decoupling is assumed to occur very rapidly while the temperature and RH changes are from observations and occur more gradually. If the decoupling was actually slower the influence of the gas-phase pump at this point in time would be reduced and the modelled decrease in NO3 (p) that occurs around 3-5 pm would be less than shown.)"

[Figure]

**Figure 3.** Evolution of the near surface atmosphere over time, taken from Wikipedia and after Stull. Period I = sunrise, Period II = growth of the daytime boundary layer, Period III = sustaining of the daytime boundary layer, Period IV = decoupling of the surface layer and residual layer near sunset.

13. There are a few (~20-30) minor typos or grammatical errors.

We have read through and done our best to identify all of these and fix them.

[revised manuscript text omitted]

**Figure S5.** The vertical distribution of the observed normalized $NO_3^-{}_{(p)}$ (dashed blue line) for the first flight leg (early morning) on 21 Jan early morning along with box model prediction of the same (blue dots). (The normalized $NO_3^-{}_{(p)} = [NO_3^-{}_{(p)}]_z/[NO_3^-{}_{(p)}]_{zmin}$, where $z$ is altitude and $z_{min}$ is lowest altitude.) Also shown are vertical profiles of temperature (yellow) and relative humidity (green) observed during the third flight leg (afternoon) on 20 Jan (dashed lines) and during the first flight leg (early morning) on 21 Jan morning (solid lines). The horizontal arrows indicate the overnight evolution of temperature and RH.

[Figure]

**Figure S6.** Diurnal profiles of (a) NO₂, (b) O₃, (c) NO and (d) the instantaneous NO₃· production rate for (red) Fresno, (blue) Parlier and (gray) Madera.

[Figure]

**Figure S7.** Relationship between the Episode 1 average vertical profiles of estimated $NO_{3\ (p)}^-$ concentrations and the night time mean wind speed. The mean wind speed is for only the nights preceding flight days. Points are colored according to altitude above ground level. The solid black line is a linear fit for altitudes < 0.45 km, with slope = -17.8 µg.s m$^{-4}$ and intercept 28.6 µg m$^{-3}$ and $r$ = -0.98. The dashed black line is a fit to all points below 1 km ($r$ = -0.96).

[Figure]

**Figure S8.** (A-D) The observed vertical profiles of $NO_3^-{}_{(g+p)}$ (black squares) from the TD-LIF and $NO_3^-{}_{(p)}$ (blue circles) for the first flight leg, along with the $NO_3^-{}_{(p)}$ for the second flight leg (purple circles). The horizontal dashed grey lines indicate the ML height at the time of the Fresno profile during flight leg 2. (E-H) The diurnal variation in the observed (blue) and modeled (green) surface-level $NO_3^-{}_{(p)}$ for each flight day in Episode 1. The temporal variation in the BLH (grey shaded area) is shown for reference.

[Figure]

**Figure S9.** Time-series of observed (top-to-bottom) $PM_1$ and particulate $NO_3^-$ concentrations, solar radiation (SRD) and temperature, $O_3$ and $O_x$ concentrations, NO, $NO_2$ and $NO_x$ concentrations, and CO concentrations with the instantaneous nitrate radical production rate, calculated as $PNO_3 = k_{NO3}[NO_2][O_3]$.

[Figure]

**Figure S10.** Ground observations of $NO_2$ (brown triangles) and temperature (green line) and the estimated OH (orange circles) and boundary layer height (gray) that are used to as inputs to the mixing model for each of the four flight days in Episode 1.

[Figure]

**Figure S11.** (left) Example model results of the influence of gas-phase HNO₃ deposition on NO₃⁻$_{(p)}$ concentrations for different assumed gas-phase nitrate fractions (indicated by color). Here, a constant $v_d = 7$ cm s$^{-1}$ and mixed-layer height of 400 m are used, and the gas and particles are assumed to remain in equilibrium at all times. The initial NO₃⁻$_{(p)}$ concentration is 10 μg m$^{-3}$. For Fresno, the observed daytime gas-phase nitrate fractions are <10%. (right) The corresponding instantaneous NO₃⁻$_{(p)}$ loss rate, in percent. The loss rate is independent of the assumed initial NO₃⁻$_{(p)}$ concentration.

[Figure]

**Figure S12.** (top row) Vertical profiles of estimated $NO_{3(p)}^{-}$ concentrations during the flight days in the second episode. The different curves are for individual flight legs. (bottom row) The individual day diurnal variability in the surface $NO_{3(p)}^{-}$ concentrations for each flight day.

[Figure]

**Figure S13.** (a) Vertical profile of the average night time (19:00-07:00) horizontal winds over Visalia, CA (65 km SE of Fresno) and the surface (10 m) wind in Fresno during Episode 2 (Jan. 29-Feb. 4). The length of the arrows corresponds to the wind speed and the direction to the average wind direction. (b) Corresponding wind roses for (b1) the surface, (b2) 125-175 m, (b3) 225-345 m, and (b4) 400-500 m. The length of each arc corresponds to the normalized probability and the colors indicate the wind speed (m/s; see legend). Data are from the National Oceanic and Atmospheric Administration, Earth System Research Laboratory, Physical Sciences Division Data and Image Archive (https://www.esrl.noaa.gov/psd/data/obs/datadisplay/, accessed 3 June 2017).

[Figure]

**Figure S14.** Comparison of model simulations of the averaged diurnal variability in surface-level $NO_{3\ (p)}^-$ between when the $HNO_3$ deposition velocity is assumed constant at 7 cm s$^{-1}$ (blue dashed line) and when the deposition velocity varies linearly with surface wind speed (solid green line). The wind-dependent $HNO_3$ deposition velocity is shown for reference (gray line).

**Table S1.** Summary of initial conditions measured at the surface-level (3 pm) used for calculation of $k_{N2O5}$ and $\gamma_{N2O5}$ for flight days during Episode 1.

| Dates | NO (ppbv) | NO$_2$ (ppbv) | O$_3$ (ppbv) | T (K) | RH (%) | NO$_3^-$$_{(p)}$ (µg m$^{-3}$) | SO$_4^{-2}$$_{(p)}$ (µg m$^{-3}$) | Cl$^-$$_{(p)}$ (µg m$^{-3}$) | $P$NO$_3^-$$_{(p)}$ (µg m$^{-3}$ nt$^{-1}$)$^\wedge$ | S$_a$ (µm$^2$ cm$^{-3}$) | $k_{N2O5}$ 1E-5 (s$^{-1}$) | $\gamma_{N2O5}$ 1E-4 |
|---|---|---|---|---|---|---|---|---|---|---|---|---|
| 17$^{th}$ – 18$^{th}$ Jan | 6.3 | 23.8 | 23.7 | 290 | 31.8 | 8.5 | 0.70 | 0.12* | 14.9 | 525.6 | 1.6 | 4.76 |
| 19$^{th}$ – 20$^{th}$ Jan | 3.7 | 21.3 | 31.5 | 290 | 36.4 | 14.3 | 0.93 | 0.35 | 14.4 | 826.5 | 1.3 | 2.46 |
| 20$^{th}$ – 21$^{st}$ Jan | 2.8 | 15.4 | 31.3 | 290 | 37.9 | 11.5 | 0.90 | 0.12 | 10.7 | 515.5 | 1.3 | 3.94 |
| 21$^{st}$ -22$^{nd}$ Jan | 1.5 | 13.3 | 41.7 | 292 | 30.0 | 9.9 | 1.0 | 0.01 | 25.3 | 295.1 | 5.1 | 2.70 |

\* Equal to 1.24 x AMS Cl
$^\wedge$ Overnight particulate nitrate production rate estimated from the difference in the maximum [NO$_3^-$$_{(p)}$] in the early-morning vertical profile at ~9:30 am and the ground-level [NO$_3^-$$_{(p)}$] the previous day at 3 pm. The notation nt$^{-1}$ indicates per night.

**Table S2**. VOC concentrations and reactivity with the $NO_3$ radical.

| VOC | Daytime concentration[a] (ppb) | $k_{rxn}$[b] ($cm^3$ molecules$^{-1}$ s$^{-1}$) | Reactivity (s$^{-1}$) |
|---|---|---|---|
| α-Pinene | 0.06 | 6.20E-12 | 9.34E-03 |
| β-Pinene | 0.02 | 2.60E-12 | 1.21E-03 |
| i-Butene | 0.11 | 3.50E-13 | 9.78E-04 |
| Isoprene | 0.05 | 7.00E-13 | 8.64E-04 |
| DMS[c] | 0.01 | 1.10E-12 | 2.86E-04 |
| trans-2-Butene | 0.03 | 3.50E-13 | 3.00E-04 |
| cis-2-Butene | 0.03 | 3.50E-13 | 2.89E-04 |
| Ethanol[c] | 2.45 | 2.00E-15 | 1.22E-04 |
| Acetaldehyde[c] | 5.14 | 2.60E-15 | 3.34E-04 |
| 1-3-Butadiene[c] | 0.03 | 1.10E-13 | 9.17E-05 |
| Propene | 0.40 | 9.40E-15 | 9.29E-05 |
| Methanol[c] | 8.52 | 1.30E-16 | 2.77E-05 |
| 1-Butene | 0.08 | 1.30E-14 | 2.54E-05 |
| m-Xylene | 0.28 | 2.30E-15 | 1.64E-05 |
| o-Xylene | 0.20 | 3.90E-15 | 1.90E-05 |
| 1-Pentene | 0.03 | 1.50E-14 | 1.04E-05 |
| Propane[c] | 4.55 | 7.00E-17 | 7.97E-06 |
| Ethene | 1.73 | 2.00E-16 | 8.65E-06 |
| 1-2-4-Trimethylbenzene | 0.09 | 1.72E-15 | 3.81E-06 |
| Ethyne | 1.81 | 1.00E-16 | 4.53E-06 |
| Others[c] | 0.29 | 2.1E-17 | 1.52E-07 |

[a]From canister samples, averaged for the afternoon period
[b]From Calvert et al., *The Mechanisms of Reactions Influencing Atmospheric Ozone*, Oxford University Press, 2015, pp. 130-160.
[c]These VOCs are assumed to react with $NO_3$ radicals to form $HNO_3$.